# MINIMUM WIDTH FOR UNIVERSAL APPROXIMATION

**Sejun Park**[†]   **Chulhee Yun**[‡]   **Jaeho Lee**[†*]   **Jinwoo Shin**[†*]
[†]KAIST AI    [‡]MIT EECS    [*]KAIST EE

## ABSTRACT

The universal approximation property of width-bounded networks has been studied as a dual of classical universal approximation results on depth-bounded networks. However, the critical width enabling the universal approximation has not been exactly characterized in terms of the input dimension $d_x$ and the output dimension $d_y$. In this work, we provide the first definitive result in this direction for networks using the RELU activation functions: The minimum width required for the universal approximation of the $L^p$ functions is exactly $\max\{d_x + 1, d_y\}$. We also prove that the same conclusion does not hold for the uniform approximation with RELU, but does hold with an additional threshold activation function. Our proof technique can be also used to derive a tighter upper bound on the minimum width required for the universal approximation using networks with general activation functions.

## 1   INTRODUCTION

The study of the expressive power of neural networks investigates what class of functions neural networks can/cannot represent or approximate. Classical results in this field are mostly focused on shallow neural networks. An example of such results is the universal approximation theorem (Cybenko, 1989; Hornik et al., 1989; Pinkus, 1999), which shows that a neural network with fixed depth and arbitrary width can approximate any continuous function on a compact set, up to arbitrary accuracy, if the activation function is continuous and nonpolynomial. Another line of research studies the memory capacity of neural networks (Baum, 1988; Huang and Babri, 1998; Huang, 2003), trying to characterize the maximum number of data points that a given neural network can memorize.

After the advent of deep learning, researchers started to investigate the benefit of depth in the expressive power of neural networks, in an attempt to understand the success of deep neural networks. This has led to interesting results showing the existence of functions that require the network to be extremely wide for shallow networks to approximate, while being easily approximated by deep and narrow networks (Telgarsky, 2016; Eldan and Shamir, 2016; Lin et al., 2017; Poggio et al., 2017). A similar trade-off between depth and width in expressive power is also observed in the study of the memory capacity of neural networks (Yun et al., 2019; Vershynin, 2020).

In search of a deeper understanding of the depth in neural networks, a *dual* scenario of the classical universal approximation theorem has also been studied (Lu et al., 2017; Hanin and Sellke, 2017; Johnson, 2019; Kidger and Lyons, 2020). Instead of bounded depth and arbitrary width studied in classical results, the dual problem studies whether universal approximation is possible with a network of *bounded width and arbitrary depth*. A very interesting characteristic of this setting is that there exists a *critical threshold* on the width that allows a neural network to be a universal approximator. For example, one of the first results (Lu et al., 2017) in the literature shows that universal approximation of $L^1$ functions from $\mathbb{R}^{d_x}$ to $\mathbb{R}$ is possible for a width-$(d_x + 4)$ RELU network, but *impossible* for a width-$d_x$ RELU network. This implies that the minimum width required for universal approximation lies between $d_x + 1$ and $d_x + 4$. Subsequent results have shown upper/lower bounds on the minimum width, but none of the results has succeeded in a tight characterization of the minimum width.

### 1.1   WHAT IS KNOWN SO FAR?

Before summarizing existing results, we first define function classes studied in the literature. For a domain $\mathcal{X} \subseteq \mathbb{R}^{d_x}$ and a codomain $\mathcal{Y} \subseteq \mathbb{R}^{d_y}$, we define $C(\mathcal{X}, \mathcal{Y})$ to be the class of continuous

---

[*]emails: sejun.park@kaist.ac.kr, chulheey@mit.edu, jaeho-lee@kaist.ac.kr, jinwoos@kaist.ac.kr

Table 1: A summary of known upper/lower bounds on minimum width for universal approximation. In the table, $\mathcal{K} \subset \mathbb{R}^{d_x}$ denotes a compact domain, and $p \in [1, \infty)$. "Conti." is short for continuous.

| Reference | Function class | Activation $\rho$ | Upper/lower bounds |
|---|---|---|---|
| Lu et al. (2017) | $L^1(\mathbb{R}^{d_x}, \mathbb{R})$ | RELU | $d_x + 1 \leq w_{\min} \leq d_x + 4$ |
| | $L^1(\mathcal{K}, \mathbb{R})$ | RELU | $w_{\min} \geq d_x$ |
| Hanin and Sellke (2017) | $C(\mathcal{K}, \mathbb{R}^{d_y})$ | RELU | $d_x + 1 \leq w_{\min} \leq d_x + d_y$ |
| Johnson (2019) | $C(\mathcal{K}, \mathbb{R})$ | uniformly conti.[†] | $w_{\min} \geq d_x + 1$ |
| Kidger and Lyons (2020) | $C(\mathcal{K}, \mathbb{R}^{d_y})$ | conti. nonpoly[‡] | $w_{\min} \leq d_x + d_y + 1$ |
| | $C(\mathcal{K}, \mathbb{R}^{d_y})$ | nonaffine poly | $w_{\min} \leq d_x + d_y + 2$ |
| | $L^p(\mathbb{R}^{d_x}, \mathbb{R}^{d_y})$ | RELU | $w_{\min} \leq d_x + d_y + 1$ |
| **Ours** (Theorem 1) | $L^p(\mathbb{R}^{d_x}, \mathbb{R}^{d_y})$ | RELU | $w_{\min} = \max\{d_x + 1, d_y\}$ |
| **Ours** (Theorem 2) | $C([0,1], \mathbb{R}^2)$ | RELU | $w_{\min} = 3 > \max\{d_x + 1, d_y\}$ |
| **Ours** (Theorem 3) | $C(\mathcal{K}, \mathbb{R}^{d_y})$ | RELU+STEP | $w_{\min} = \max\{d_x + 1, d_y\}$ |
| **Ours** (Theorem 4) | $L^p(\mathcal{K}, \mathbb{R}^{d_y})$ | conti. nonpoly[‡] | $w_{\min} \leq \max\{d_x + 2, d_y + 1\}$ |

[†] requires that $\rho$ is uniformly approximated by a sequence of one-to-one functions.
[‡] requires that $\rho$ is continuously differentiable at some $z$ with $\rho'(z) \neq 0$.

functions from $\mathcal{X}$ to $\mathcal{Y}$, endowed with the uniform norm: $\|f\|_\infty := \sup_{x \in \mathcal{X}} \|f(x)\|_\infty$. For $p \in [1, \infty)$, we also define $L^p(\mathcal{X}, \mathcal{Y})$ to be the class of $L^p$ functions from $\mathcal{X}$ to $\mathcal{Y}$, endowed with the $L^p$-norm: $\|f\|_p := (\int_\mathcal{X} \|f(x)\|_p^p \, dx)^{1/p}$. The summary of known upper and lower bounds in the literature, as well as our own results, is presented in Table 1. We use $w_{\min}$ to denote the minimum width for universal approximation.

**First progress.** As aforementioned, Lu et al. (2017) show that universal approximation of $L^1(\mathbb{R}^{d_x}, \mathbb{R})$ is possible for a width-$(d_x + 4)$ RELU network, but impossible for a width-$d_x$ RELU network. These results translate into bounds on the minimum width: $d_x + 1 \leq w_{\min} \leq d_x + 4$. Hanin and Sellke (2017) consider approximation of $C(\mathcal{K}, \mathbb{R}^{d_y})$, where $\mathcal{K} \subset \mathbb{R}^{d_x}$ is compact. They prove that RELU networks of width $d_x + d_y$ are dense in $C(\mathcal{K}, \mathbb{R}^{d_y})$, while width-$d_x$ RELU networks are *not*. Although this result fully characterizes $w_{\min}$ in case of $d_y = 1$, it fails to do so for $d_y > 1$.

**General activations.** Later, extensions to activation functions other than RELU have appeared in the literature. Johnson (2019) shows that if the activation function $\rho$ is uniformly continuous and can be uniformly approximated by a sequence of one-to-one functions, a width-$d_x$ network cannot universally approximate $C(\mathcal{K}, \mathbb{R})$. Kidger and Lyons (2020) show that if $\rho$ is continuous, nonpolynomial, and continuously differentiable at some $z$ with $\rho'(z) \neq 0$, then networks of width $d_x + d_y + 1$ with activation $\rho$ are dense in $C(\mathcal{K}, \mathbb{R}^{d_y})$. Furthermore, Kidger and Lyons (2020) prove that RELU networks of width $d_x + d_y + 1$ are dense in $L^p(\mathbb{R}^{d_x}, \mathbb{R}^{d_y})$.

**Limitations of prior arts.** Note that none of the existing works succeeds in closing the gap between the upper bound (at least $d_x + d_y$) and the lower bound (at most $d_x + 1$). This gap is significant especially for applications with high-dimensional codomains (i.e., large $d_y$), arising for many practical applications of neural networks, e.g., image generation (Kingma and Welling, 2013; Goodfellow et al., 2014), language modeling (Devlin et al., 2019; Liu et al., 2019), and molecule generation (Gómez-Bombarelli et al., 2018; Jin et al., 2018). In the prior arts, the main bottleneck for proving an upper bound below $d_x + d_y$ is that they maintain all $d_x$ neurons to store the input and all $d_y$ neurons to construct the function output; this means every layer already requires at least $d_x + d_y$ neurons. In addition, the proof techniques for the lower bounds only consider the input dimension $d_x$ regardless of the output dimension $d_y$.

## 1.2 SUMMARY OF RESULTS

We mainly focus on characterizing the minimum width of RELU networks for universal approximation. Nevertheless, our results are not restricted to RELU networks; they can be generalized to networks with general activation functions. Our contributions can be summarized as follows.

- Theorem 1 states that the minimum width for RELU networks to be dense in $L^p(\mathbb{R}^{d_x}, \mathbb{R}^{d_y})$ is exactly $\max\{d_x + 1, d_y\}$. This is the first result fully characterizing the minimum width of RELU networks for universal approximation. In particular, the upper bound on the minimum width is significantly smaller than the best known result $d_x + d_y + 1$ (Kidger and Lyons, 2020).

- Given the full characterization of $w_{\min}$ of RELU networks for approximating $L^p(\mathbb{R}^{d_x}, \mathbb{R}^{d_y})$, a natural question arises: Is $w_{\min}$ also the same for $C(\mathcal{K}, \mathbb{R}^{d_y})$? We prove that it is *not* the case; Theorem 2 shows that the minimum width for RELU networks to be dense in $C([0, 1], \mathbb{R}^2)$ is 3. Namely, RELU networks of width $\max\{d_x + 1, d_y\}$ are *not* dense in $C(\mathcal{K}, \mathbb{R}^{d_y})$ in general.

- In light of Theorem 2, is it impossible to approximate $C(\mathcal{K}, \mathbb{R}^{d_y})$ in general while maintaining width $\max\{d_x + 1, d_y\}$? Theorem 3 shows that an additional activation comes to rescue. We show that if networks use *both* RELU and threshold activation functions (which we refer to as STEP)[1], they can universally approximate $C(\mathcal{K}, \mathbb{R}^{d_y})$ with the minimum width $\max\{d_x + 1, d_y\}$.

- Our proof techniques for tight upper bounds are not restricted to RELU networks. In Theorem 4, we extend our results to general activation functions covered in Kidger and Lyons (2020).

### 1.3 ORGANIZATION

We first define necessary notation in Section 2. In Section 3, we formally state our main results and discuss their implications. In Section 4, we present our "coding scheme" for proving upper bounds on the minimum width in Theorems 1, 3 and 4. In Section 5, we prove the lower bound in Theorem 2 by explicitly constructing a counterexample. Finally, we conclude the paper in Section 6. We note that all formal proofs of Theorems 1–4 are presented in Appendix.

## 2 PROBLEM SETUP AND NOTATION

Throughout this paper, we consider fully-connected neural networks that can be described as an alternating composition of affine transformations and activation functions. Formally, we consider the following setup: Given a set of activation functions $\Sigma$, an $L$-layer neural network $f$ of input dimension $d_x$, output dimension $d_y$, and hidden layer dimensions $d_1, \ldots, d_{L-1}$[2] is represented as

$$f := t_L \circ \sigma_{L-1} \circ \cdots \circ t_2 \circ \sigma_1 \circ t_1, \tag{1}$$

where $t_\ell : \mathbb{R}^{d_{\ell-1}} \to \mathbb{R}^{d_\ell}$ is an affine transformation and $\sigma_\ell$ is a vector of activation functions:

$$\sigma_\ell(x_1, \ldots, x_{d_\ell}) = \big(\rho_1(x_1), \ldots, \rho_{d_\ell}(x_{d_\ell})\big),$$

where $\rho_i \in \Sigma$. While we mostly consider the cases where $\Sigma$ is a singleton (e.g., $\Sigma = \{\text{RELU}\}$), we also consider the case where $\Sigma$ contains both RELU and STEP activation functions as in Theorem 3. We denote a neural network with $\Sigma = \{\rho\}$ by a "$\rho$ network" and a neural network with $\Sigma = \{\rho_1, \rho_2\}$ by a "$\rho_1+\rho_2$ network." We define the *width* $w$ of $f$ as the maximum over $d_1, \ldots, d_{L-1}$. We use "$\rho$ networks (or $\rho_1 + \rho_2$ networks) of width $w$" for denoting the collection of all $\rho$ networks (or $\rho_1 + \rho_2$ networks) of width $w$ having a finite number of layers.

For describing the universal approximation of neural networks, we say $\rho$ networks (or $\rho_1+\rho_2$ networks) of width $w$ are dense in $C(\mathcal{X}, \mathcal{Y})$ if for any $f^* \in C(\mathcal{X}, \mathcal{Y})$ and $\varepsilon > 0$, there exists a $\rho$ network (or a $\rho_1+\rho_2$ network) $f$ of width $w$ such that $\|f^* - f\|_\infty \leq \varepsilon$. Likewise, we say $\rho$ networks (or $\rho_1+\rho_2$ networks) are dense in $L^p(\mathcal{X}, \mathcal{Y})$ if for any $f^* \in L^p(\mathcal{X}, \mathcal{Y})$ and $\varepsilon > 0$, there exists a $\rho$ network (or a $\rho_1+\rho_2$ network) $f$ such that $\|f^* - f\|_p \leq \varepsilon$.

## 3 MINIMUM WIDTH FOR UNIVERSAL APPROXIMATION

$L^p$ **approximation with RELU.** We present our main theorems in this section. First, for universal approximation of $L^p(\mathbb{R}^{d_x}, \mathbb{R}^{d_y})$ using RELU networks, we give the following theorem.

**Theorem 1.** *For any $p \in [1, \infty)$, RELU networks of width $w$ are dense in $L^p(\mathbb{R}^{d_x}, \mathbb{R}^{d_y})$ if and only if $w \geq \max\{d_x + 1, d_y\}$.*

---

[1]The threshold activation function (i.e., STEP) denotes $x \mapsto \mathbf{1}[x \geq 0]$.

[2]For simplicity of notation, we let $d_0 := d_x$ and $d_L := d_y$.

This theorem shows that the minimum width $w_{\min}$ for universal approximation is exactly equal to $\max\{d_x + 1, d_y\}$. In order to provide a tight characterization of $w_{\min}$, we show three new upper and lower bounds: $w_{\min} \leq \max\{d_x + 1, d_y\}$ through a construction utilizing a coding approach, $w_{\min} \geq d_y$ through a volumetric argument, and $w_{\min} \geq d_x + 1$ through an extension of the same lower bound for $L^1(\mathbb{R}^{d_x}, \mathbb{R}^{d_y})$ (Lu et al., 2017). Combining these bounds gives the tight minimum width $w_{\min} = \max\{d_x + 1, d_y\}$.

Notably, using our new proof technique, we overcome the limitation of existing upper bounds that require width at least $d_x + d_y$. Our construction first encodes the $d_x$ dimensional input vectors into one-dimensional codewords, and maps the codewords to target codewords using memorization, and decodes the target codewords to $d_y$ dimensional output vectors. Since we construct the map from input to target using scalar codewords, we bypass the need to use $d_x + d_y$ hidden nodes. More details are found in Section 4. Proofs of the lower bounds are deferred to Appendix B.

**Uniform approximation with RELU.** In Theorem 1, we have seen the tight characterization $w_{\min} = \max\{d_x + 1, d_y\}$ for $L^p(\mathbb{R}^{d_x}, \mathbb{R}^{d_y})$ functions. Does the same hold for $C(\mathcal{K}, \mathbb{R}^{d_y})$, for a compact $\mathcal{K} \subset \mathbb{R}^{d_x}$? Surprisingly, we show that the same conclusion does *not* hold in general. Indeed, we show the following result, proving that width $\max\{d_x + 1, d_y\}$ is *provably insufficient* for $d_x = 1, d_y = 2$.

**Theorem 2.** RELU *networks of width $w$ are dense in $C([0, 1], \mathbb{R}^2)$ if and only if $w \geq 3$.*

Theorem 2 translates to $w_{\min} = 3$, and the upper bound $w_{\min} \leq 3 = d_x + d_y$ is given by Hanin and Sellke (2017). The key is to prove a lower bound $w_{\min} \geq 3$, i.e., width 2 is not sufficient. Recall from Section 1.1 that all the known lower bounds are limited to showing that width $d_x$ is insufficient for universal approximation. A closer look at their proof techniques reveals that they heavily rely on the fact that the hidden layers have the same dimensions as the input space. As long as the width $w > d_x$, their arguments break because such a network maps the input space into a *higher-dimensional* space.

Although only for $d_x = 1$ and $d_y = 2$, we overcome this limitation of the prior arts and show that width $w = 2 > d_x$ is *insufficient* for universal approximation, by providing a counterexample. We use a novel topological argument which comes from a careful observation on the image created by RELU operations. In particular, we utilize the property of RELU that it projects all negative inputs to zero, without modifying any positive inputs. We believe that our proof will be of interest to readers and inspire follow-up works. Please see Section 5 for more details.

Theorem 1 and Theorem 2 together imply that for RELU networks, approximating $C(\mathcal{K}, \mathbb{R}^{d_y})$ requires more width than approximating $L^p(\mathbb{R}^{d_x}, \mathbb{R}^{d_y})$. Interestingly, this is in stark contrast with existing results, where the minimum *depth* of RELU networks for approximating $C(\mathcal{K}, \mathbb{R}^{d_y})$ is two (Leshno et al., 1993) but it is greater than two for approximating $L^p(\mathbb{R}^{d_x}, \mathbb{R}^{d_y})$ (Wang and Qu, 2019).

**Uniform approximation with RELU+STEP.** While width $\max\{d_x + 1, d_y\}$ is insufficient for RELU networks to be dense in $C(\mathcal{K}, \mathbb{R}^{d_y})$, an additional STEP activation function helps achieve the minimum width $\max\{d_x + 1, d_y\}$, as stated in the theorem below.

**Theorem 3.** RELU+STEP *networks of width $w$ are dense in $C(\mathcal{K}, \mathbb{R}^{d_y})$ if and only if $w \geq \max\{d_x + 1, d_y\}$.*

Theorem 2 and Theorem 3 indicate that the minimum width for universal approximation is indeed *dependent* on the choice of activation functions. This is also in contrast to the classical results where RELU networks of depth 2 are universal approximators (Leshno et al., 1993), i.e., the minimum depths for universal approximation are identical for both RELU networks and RELU+STEP networks.

Theorem 3 comes from a similar proof technique as Theorem 1. Due to its discontinuous nature, the STEP activation can be used in our encoder to quantize the input without introducing uniform norm errors. Lower bounds on $w_{\min}$ can be proved in a similar way as Theorem 1.

**General activations.** Our proof technique for upper bounds in Theorems 1 and 3 can be easily extended to networks using general activations. Indeed, we prove the following theorem, which shows that adding a width of 1 is enough to cover the networks with general activations.

**Theorem 4.** *Let $\rho : \mathbb{R} \rightarrow \mathbb{R}$ be any continuous nonpolynomial function which is continuously differentiable at some $z$ with $\rho'(z) \neq 0$. Then, $\rho$ networks of width $w$ are dense in $L^p(\mathcal{K}, \mathbb{R}^{d_y})$ for all $p \in [1, \infty)$ if $w \geq \max\{d_x + 2, d_y + 1\}$.*

Figure 1: Illustration of the coding scheme

Please notice that unlike other theorems, Theorem 4 only proves an upper bound $w_{\min} \leq \max\{d_x + 2, d_y + 1\}$. We note that Theorem 4 significantly improves over the previous upper bound of width $d_x + d_y + 1$ by (Kidger and Lyons, 2020, Remark 4.10).

## 4    TIGHT UPPER BOUND ON MINIMUM WIDTH

In this section, we present the main idea for constructing networks achieving the minimum width for universal approximation, and then sketch the proofs of upper bounds in Theorems 1, 3, and 4.

### 4.1    CODING SCHEME FOR UNIVERSAL APPROXIMATION

We now illustrate the main idea underlying the construction of neural networks that achieve the minimum width. To this end, we consider an approximation of a target continuous function $f^* \in C([0,1]^{d_x}, [0,1]^{d_y})$; however, our main idea can be easily generalized to other domain, codomain, and $L^p$ functions. Our construction can be viewed as a *coding scheme* in essence, consisting of three parts: *encoder, memorizer, and decoder*. First, the encoder encodes an input vector $x$ to a one-dimensional codeword. Then, the memorizer maps the codeword to a one-dimensional target codeword that is encoded with respect to the corresponding target $f^*(x)$. Finally, the decoder maps the target codeword to a target vector which is sufficiently close to $f^*(x)$. Note that one can view the encoder, memorizer, and decoder as functions mapping from $d_x$-dimension to 1-dimension, then to 1-dimension, and finally to $d_y$-dimension.

The spirit of the coding scheme is that the three functions can be constructed using the idea of the prior results such as (Hanin and Sellke, 2017). Recall that Hanin and Sellke (2017) approximate any continuous function mapping $n$-dimensional inputs to $m$-dimensional outputs using RELU networks of width $n + m$. Under this intuition, we construct the encoder, the memorizer, and the decoder by RELU+STEP networks (or RELU networks) of width $d_x + 1, 2, d_y$, respectively; these constructions result in the tight upper bound $\max\{d_x + 1, d_y\}$. Here, the decoder requires width $d_y$ instead of $d_y + 1$, as we only construct the first $d_y - 1$ coordinates of the output, and recover the last output coordinate from a linear combination of the target codeword and the first $d_y - 1$ coordinates. Note that we construct the exact encoder, decoder, and memorizer as prior results only approximate them.

Next, we describe the operation of each part. We explain their neural network constructions in subsequent subsections.

**Encoder.**    Before introducing the encoder, we first define a quantization function $q_n : [0,1] \rightarrow \mathcal{C}_n$ for $n \in \mathbb{N}$ and $\mathcal{C}_n := \{0, 2^{-n}, 2 \times 2^{-n}, \ldots, 1 - 2^{-n}\}$ as

$$q_n(x) := \max\{c \in \mathcal{C}_n : c \leq x\}.$$

In other words, given any $x \in [0,1)$, $q_n(x)$ preserves the first $n$ bits in the binary representation of $x$ and discards the rest; $x = 1$ is mapped to $1 - 2^{-n}$. Note that the error from the quantization is always less than or equal to $2^{-n}$.

The encoder encodes each input $x \in [0,1]^{d_x}$ to some scalar value via the function $\texttt{encode}_K : \mathbb{R}^{d_x} \rightarrow \mathcal{C}_{d_x K}$ for some $K \in \mathbb{N}$ defined as

$$\texttt{encode}_K(x) := \sum_{i=1}^{d_x} q_K(x_i) \times 2^{-(i-1)K}.$$

In other words, $\texttt{encode}_K(x)$ quantizes each coordinate of $x$ by a $K$-bit binary representation and concatenates the quantized coordinates into a single scalar value having a $(d_x K)$-bit binary

representation. Note that if one "decodes" a codeword $\texttt{encode}_K(x)$ back to a vector $\hat{x}$ as[3]

$$\{\hat{x}\} := \left(\texttt{encode}_K^{-1} \circ \texttt{encode}_K(x)\right) \cap \mathcal{C}_K^{d_x},$$

then $\|x - \hat{x}\|_\infty \leq 2^{-K}$. Namely, the "information loss" incurred by the encoding can be made arbitrarily small by choosing large $K$.

**Memorizer.**  The memorizer maps each codeword $\texttt{encode}_K(x) \in \mathcal{C}_{d_x K}$ to its target codeword via the function $\texttt{memorize}_{K,M} : \mathcal{C}_{d_x K} \to \mathcal{C}_{d_y M}$ for some $M \in \mathbb{N}$, defined as

$$\texttt{memorize}_{K,M}\left(\texttt{encode}_K(x)\right) := \texttt{encode}_M\left(f^* \circ q_K(x)\right)$$

where $q_K$ is applied coordinate-wise for a vector. We note that $\texttt{memorize}_{K,M}$ is well-defined as each $\texttt{encode}_K(x) \in \mathcal{C}_{d_x K}$ corresponds to a unique $q_K(x) \in \mathcal{C}_K^{d_x}$. Here, one can observe that the target of the memorizer contains the information of the target value since $\texttt{encode}_M\left(f^* \circ q_K(x)\right)$ contains information of $f^*$ at a quantized version of $x$, and the information loss due to quantization can be made arbitrarily small by choosing large enough $K$ and $M$.

**Decoder.**  The decoder decodes each codeword generated by the memorizer using the function $\texttt{decode}_M : \mathcal{C}_{d_y M} \to \mathcal{C}_M^{d_y}$ defined as

$$\texttt{decode}_M(c) := \hat{x} \quad \text{where} \quad \{\hat{x}\} := \texttt{encode}_M^{-1}(c) \cap \mathcal{C}_M^{d_y}.$$

Combining $\texttt{encode}$, $\texttt{memorize}$, and $\texttt{decode}$ completes our coding scheme for approximating $f^*$. One can observe that our coding scheme is equivalent to $q_M \circ f^* \circ q_K$ which can approximate the target function $f^*$ within any $\varepsilon > 0$ error, i.e.,

$$\sup_{x \in [0,1]^{d_x}} \|f^*(x) - \texttt{decode}_M \circ \texttt{memorize}_{K,M} \circ \texttt{encode}_K(x)\|_\infty \leq \varepsilon$$

by choosing large enough $K, M \in \mathbb{N}$ so that $\omega_{f^*}(2^{-K}) + 2^{-M} \leq \varepsilon$.[4]

In the remainder of this section, we discuss how each part of the coding scheme can be implemented with a neural network using RELU+STEP activations (Section 4.2), RELU activation (Section 4.3), and other general activations (Section 4.4).

## 4.2   TIGHT UPPER BOUND ON MINIMUM WIDTH OF RELU+STEP NETWORKS (THEOREM 3)

In this section, we discuss how we explicitly construct our coding scheme to approximate functions in $C(\mathcal{K}, \mathbb{R}^{d_y})$ using a width-$(\max\{d_x + 1, d_y\})$ RELU+STEP network. This results in the tight upper bound in Theorem 3.

First, the encoder consists of quantization functions $q_K$ and a linear transformation. However, as $q_K$ is discontinuous and cannot be uniformly approximated by any continuous function, we utilize the discontinuous STEP activation to exactly construct the encoder via a RELU+STEP network of width $d_x + 1$. On the other hand, the memorizer and the decoder maps a finite number of scalar values (i.e., $\mathcal{C}_{d_x K}$ and $\mathcal{C}_{d_y M}$, respectively) to their target values/vectors. Such maps can be easily implemented by piecewise linear functions; hence, they can be exactly constructed by RELU networks of width 2 and $d_y$, respectively, as discussed in Section 4.1. Note that STEP is used only for constructing the encoder.

In summary, all parts of our coding scheme can be *exactly* constructed by RELU+STEP networks of width $d_x + 1$, 2, and $d_y$. Thus, the overall RELU+STEP network has width $\max\{d_x + 1, d_y\}$. Furthermore, it can approximate the target continuous function $f^*$ within arbitrary uniform error by choosing sufficiently large $K$ and $M$.

## 4.3   TIGHT UPPER BOUND ON MINIMUM WIDTH OF RELU NETWORKS (THEOREM 1)

The construction of width-$(\max\{d_x + 1, d_y\})$ RELU network for approximating $L^p(\mathbb{R}^{d_x}, \mathbb{R}^{d_y})$ (i.e., the tight upper bound in Theorem 1) is almost identical to the RELU+STEP network construction

---

[3]Here, $\texttt{encode}_K^{-1}$ denotes the preimage of $\texttt{encode}_K$ and $\mathcal{C}_K^{d_x}$ is the Cartesian product of $d_x$ copies of $\mathcal{C}_K$.
[4]$\omega_{f^*}$ denotes the modulus of continuity of $f^*$: $\|f^*(x) - f^*(x')\|_\infty \leq \omega_{f^*}(\|x - x'\|_\infty) \ \forall x, x' \in [0,1]^{d_x}$.

in Section 4.2. Since any $L^p$ function can be approximated by a continuous function with compact support, we aim to approximate continuous $f^* : [0,1]^{d_x} \to [0,1]^{d_y}$ here as in our coding scheme.

Since the memorizer and the decoder can be exactly constructed by RELU networks, we only discuss the encoder here. As we discussed in the last section, the encoder cannot be uniformly approximated by continuous functions (i.e., RELU networks). Nevertheless, it can be implemented by continuous functions except for a subset of the domain around the discontinuities, and this subset can be made arbitrarily small in terms of the Lebesgue measure. That is, we construct the encoder using a RELU network of width $d_x + 1$ for $[0,1]^{d_x}$ except for a small subset, which enables us to approximate the encoder in the $L^p$-norm. Combining with the memorizer and the decoder, we obtain a RELU network of width $\max\{d_x + 1, d_y\}$ that approximates the target function $f^*$ in the $L^p$-norm.

## 4.4 TIGHTENING UPPER BOUND ON MINIMUM WIDTH OF GENERAL NETWORKS (THEOREM 4)

Our network construction can be generalized to general activation functions using existing results on approximation of $C(\mathcal{K}, \mathbb{R}^{d_y})$ functions. For example, Kidger and Lyons (2020) show that if the activation $\rho$ is continuous, nonpolynomial, and continuously differentiable at some $z$ with $\rho'(z) \neq 0$, then $\rho$ networks of width $d_x + d_y + 1$ are dense in $C(\mathcal{K}, \mathbb{R}^{d_y})$. Applying this result to our encoder, memorizer, and decoder constructions of RELU networks, it follows that if $\rho$ satisfies the conditions above, then $\rho$ networks of width $\max\{d_x + 2, d_y + 1\}$ are dense in $L^p(\mathcal{K}, \mathbb{R}^{d_y})$, i.e., Theorem 4 holds. We note that any universal approximation result for $C(\mathcal{K}, \mathbb{R}^{d_y})$ by networks using other activation functions, other than Kidger and Lyons (2020), can also be combined with our construction.

## 4.5 NUMBER OF LAYERS FOR MINIMUM WIDTH UNIVERSAL APPROXIMATORS

Lastly, we discuss the required number of layers for our network constructions achieving the tight minimum width in Theorem 1 and 3. First, our RELU + STEP network construction of width $\max\{d_x + 1, d_y\}$ for approximating $f^* \in C([0,1]^{d_x}, \mathbb{R}^{d_y})$ requires $O(2^{d_x \cdot K} + 2^M)$ layers. Here, $O(2^K)$ layers are for the encoder, $O(2^{d_x \cdot K})$ layers are for the memorizer, and $O(2^M)$ layers for the decoder.[5] Since $K, M$ must satisfy $\omega_{f^*}(2^{-K}) + 2^{-M} \leq \varepsilon$ for approximating $f^*$ in $\varepsilon$ error (see Section 4.1), we choose $K = \lceil -\log_2(\omega_{f^*}^{-1}(\varepsilon/2)) \rceil$, $M = \lceil -\log_2(\varepsilon/2) \rceil$ so that $\omega_{f^*}(2^{-K}), 2^{-M} \leq \varepsilon/2$. Namely, the overall required number of layers (i.e., parameters) for approximating $f^*$ in $\varepsilon$ error is $O\big((\omega_{f^*}^{-1}(\varepsilon/2))^{-d_x} + 1/\varepsilon\big)$.

This number is closely related to the number of parameters for approximating arbitrary $f^* \in C([0,1]^{d_x}, \mathbb{R}^{d_y})$ in $\varepsilon$ error: For RELU networks of a constantly bounded number of layers, $\Theta\big((\omega_{f^*}^{-1}(\Omega(\varepsilon)))^{-d_x}\big)$ parameters are necessary and sufficient (Yarotsky, 2018). On the other hand, for RELU networks whose number of layers can increase with the number of parameters, $O\big((\omega_{f^*}^{-1}(\Omega(\varepsilon)))^{-d_x/2}\big)$ parameters are sufficient (Yarotsky, 2018). However, while the number of layers grows with the number of parameters in our construction, it requires $O\big((\omega_{f^*}^{-1}(\Omega(\varepsilon)))^{-d_x} + 1/\varepsilon\big)$ parameters. Namely, the number of parameters in constructions achieving the tight minimum width can be improved.

Likewise, for proving the tight upper bound in Theorem 1, for approximating $f^* \in L^p(\mathbb{R}^{d_x}, \mathbb{R}^{d_y})$ in $\varepsilon$ error, our RELU network construction also requires $O\big((\omega_f^{-1}(\Omega(\varepsilon)))^{-d_x} + 1/\varepsilon\big)$ layers where $f$ is some continuous function satisfying $\|f^* - f\|_p = \varepsilon/2$.

## 5 TIGHT LOWER BOUND ON MINIMUM WIDTH

The purpose of this section is to prove the tight lower bound in Theorem 2, i.e., there exist $f^* \in C([0,1], \mathbb{R}^2)$ and $\varepsilon > 0$ satisfying the following property: For any width-2 RELU network $f$, we have $\|f^* - f\|_\infty > \varepsilon$. Our construction of $f^*$ is based on topological properties of RELU networks, which we study in Section 5.1. Then, we introduce a counterexample $f^*$ and prove that $f^*$ cannot be approximated by width-2 RELU networks in Section 5.2.

---

[5]$d_x, d_y$ are considered as constants.

## 5.1 TOPOLOGICAL PROPERTIES OF RELU NETWORKS

We first interpret a width-2 RELU network $f : \mathbb{R} \to \mathbb{R}^2$ as below, following (1):

$$f := t_L \circ \sigma \circ \cdots \circ \sigma \circ t_2 \circ \sigma \circ t_1$$

where $L \in \mathbb{N}$ denotes the number of layers, $t_1 : \mathbb{R} \to \mathbb{R}^2$ and $t_\ell : \mathbb{R}^2 \to \mathbb{R}^2$ for $\ell > 1$ are affine transformations, and $\sigma$ is the coordinate-wise RELU. Without loss of generality, we assume that $t_\ell$ is invertible for all $\ell > 1$, as invertible affine transformations are dense in the space of affine transformations on bounded support, endowed with the uniform norm. To illustrate the topological properties of $f$ better, we reformulate $f$ as follows:

$$f = (\phi_{L-1}^{-1} \circ \sigma \circ \phi_{L-1}) \circ \cdots \circ (\phi_2^{-1} \circ \sigma \circ \phi_2) \circ (\phi_1^{-1} \circ \sigma \circ \phi_1) \circ t^\dagger \tag{2}$$

where $\phi_\ell$ and $t^\dagger$ are defined as $t^\dagger := t_L \circ \cdots \circ t_1$ and $\phi_\ell := (t_L \circ \cdots \circ t_{\ell+1})^{-1}$, i.e., $t_\ell = \phi_\ell \circ \phi_{\ell-1}^{-1}$ for $\ell \geq 2$ and $t_1 = \phi_1 \circ t^\dagger$. Under the reformulation (2), $f$ first maps inputs through an affine transformation $t^\dagger$, then it sequentially applies $\phi_\ell^{-1} \circ \sigma \circ \phi_\ell$. Here, $\phi_\ell^{-1} \circ \sigma \circ \phi_\ell$ can be viewed as changing the coordinate system using $\phi_\ell$, applying RELU in the modified coordinate system, and then returning back to the original coordinate system via $\phi_\ell^{-1}$. Under this reformulation, we present the following lemmas, whose proofs are presented in Appendices B.4–B.5.

**Lemma 5.** *Let $\phi : \mathbb{R}^2 \to \mathbb{R}^2$ be an invertible affine transformation. Then, there exist $a_1, a_2 \in \mathbb{R}^2$ and $b_1, b_2 \in \mathbb{R}$ determined by $\phi$ such that the following statements hold for $\mathcal{S} := \{x : \langle a_1, x \rangle + b_1 \geq 0, \langle a_2, x \rangle + b_2 \geq 0\}$ and $x' := \phi^{-1} \circ \sigma \circ \phi(x)$.*

- *If $x \in \mathcal{S}$, then $x' = x$.*

- *If $x \in \mathbb{R}^2 \setminus \mathcal{S}$, then $x' \neq x$ and $x' \in \partial \mathcal{S}$.[6]*

**Lemma 6.** *Let $\phi : \mathbb{R}^2 \to \mathbb{R}^2$ be an invertible affine transformation. Suppose that $x$ is in a bounded path-component[7] of $\mathbb{R}^2 \setminus \mathcal{T}$ for some $\mathcal{T} \subset \mathbb{R}^2$. Then, the following statements hold for $x' := \phi^{-1} \circ \sigma \circ \phi(x)$ and $\mathcal{T}' := \phi^{-1} \circ \sigma \circ \phi(\mathcal{T})$.*

- *If $x' = x$ and $x' \notin \mathcal{T}'$, then $x'$ is in a bounded path-component of $\mathbb{R}^2 \setminus \mathcal{T}'$.*

- *If $x' \neq x$, then $x' \in \mathcal{T}'$.*

Lemma 5 follows from the fact that output of RELU is identity to nonnegative coordinates, and is zero to negative coordinates. In particular, $a_1, b_1$ and $a_2, b_2$ in Lemma 5 correspond to the axes of the "modified" coordinate system before applying $\sigma$. Under the same property of RELU, Lemma 6 states that if a point $x$ is surrounded by a set $\mathcal{T}$, after applying $\phi^{-1} \circ \sigma \circ \phi$, either the point stays at the same position and surrounded by the image of $\mathcal{T}$ or intersects with the image of $\mathcal{T}$. Based on these observations, we are now ready to introduce our counterexample.

## 5.2 COUNTEREXAMPLE

Our counterexample $f^* : [0, 1] \to \mathbb{R}^2$ is illustrated in Figure 2(a) where $f^*([0, p_1])$ is drawn in red from $(4, 3)$ to $(0, 0)$, $f^*((p_1, p_2))$ is drawn in black from $(0, 0)$ to $(-1, 0)$, and $f^*([p_2, 1])$ is drawn in blue from $(-1, 0)$ to $(1, 0)$, for some $0 < p_1 < p_2 < 1$, e.g., $p_1 = \frac{1}{3}, p_2 = \frac{2}{3}$. In this section, we suppose for contradiction that there exists a RELU network $f$ of width 2 such that $\|f^* - f\|_\infty \leq \frac{1}{100}$. To this end, consider the mapping by the first $\ell$ layers of $f$:

$$g_\ell := (\phi_\ell^{-1} \circ \sigma \circ \phi_\ell) \circ \cdots \circ (\phi_1^{-1} \circ \sigma \circ \phi_1) \circ t^\dagger.$$

Our proof is based on the fact: If $g_\ell(x) = g_\ell(x')$, then $f(x) = f(x')$. Thus, the following must hold:

$$\text{if } \|f^* - f\|_\infty \leq \tfrac{1}{100}, \text{ then } g_\ell([0, p_1]) \cap g_\ell([p_2, 1]) = \emptyset \text{ for all } \ell \geq 1. \tag{3}$$

Let $\mathcal{B} := (-2, 2) \times (-1, 1)$ (the gray box in Figure 2(b)) and $\ell^* \in \mathbb{N}$ be the largest $\ell$ such that $\phi_\ell^{-1} \circ \sigma \circ \phi_\ell(\mathcal{B}) \neq \mathcal{B}$. This means that after the $\ell^*$-th layer, everything inside the box $\mathcal{B}$ never gets

---

[6]$\partial \mathcal{S}$ denotes the boundary set of $\mathcal{S}$.

[7]$\mathcal{S} \subset \mathcal{T}$ is a path-component of $\mathcal{T}$ if $\mathcal{S}$ is a maximal set satisfying the following condition: For any $x_1, x_2 \in \mathcal{S}$, there exists a continuous function $f : [0, 1] \to \mathcal{S}$ such that $f(0) = x_1$ and $f(1) = x_2$.

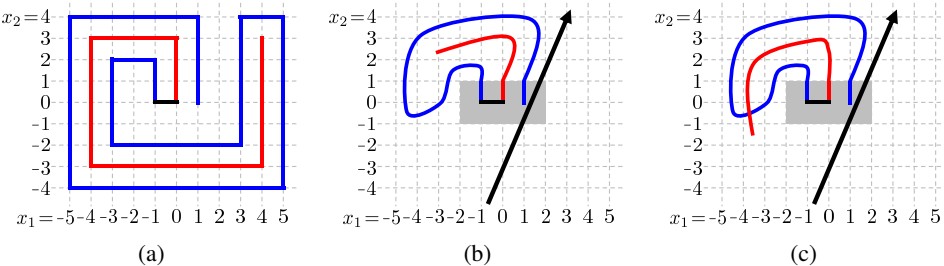

Figure 2: (a) Illustration of the image of $f^* : [0,1] \to \mathbb{R}^2$ (b, c) Examples of $g_{\ell^*}([0,1])$.

affected by RELU operations. In other words, $f([0,1]) \cap \mathcal{B}$ must have been constructed in the first $\ell^*$ layers. Under this observation, the boundary of $\mathcal{S}$ in Lemma 5 determined by $\phi_{\ell^*}$ must intersect with $\mathcal{B}$: If $\mathcal{B} \subset \mathcal{S}$, then $\phi_{\ell^*}^{-1} \circ \sigma \circ \phi_\ell(\mathcal{B}) = \mathcal{B}$ which contradicts the definition of $\ell^*$ and if $\mathcal{B} \cap \mathcal{S} = \emptyset$, then $f([0,1]) \cap \mathcal{B}$ cannot be constructed in the first $\ell^*$ layers. Therefore, there must exist a line (e.g., a line containing one boundary of $\mathcal{S}$, see the arrow in Figure 2(b)) intersecting with $\mathcal{B}$, such that the image $g_{\ell^*}([0,1])$ lies in one side of the line. Since the image of the entire network $f([0,p_1])$ is on both sides of the line, we have $g_{\ell^*}([0,p_1]) \neq f([0,p_1])$, which implies that the remaining layers $\ell^* + 1, \ldots, L-1$ must have moved the image $g_{\ell^*}([0,p_1]) \setminus \mathcal{B}$ to $f([0,p_1]) \setminus \mathcal{B}$; this also implies $g_{\ell^*}([0,p_1]) \setminus \mathcal{B} \neq \emptyset$. A similar argument gives $g_{\ell^*}([p_2,1]) \setminus \mathcal{B} \neq \emptyset$.

Since $f([0,1]) \cap \mathcal{B}$ must have been constructed in the first $\ell^*$ layers, as illustrated in Figures 2(b) and 2(c), the boundary $\partial \mathcal{B}$ intersects with $g_{\ell^*}([p_2,1])$ (the blue line) near points $(-1,1)$ and $(1,1)$. Hence, $\mathcal{T} := g_{\ell^*}([p_2,1]) \cup \mathcal{B}$ forms a "closed loop." Also, $\partial \mathcal{B}$ intersects with $g_{\ell^*}([0,p_1])$ near the point $(0,1)$, so there must exist a point in $g_{\ell^*}([0,p_1]) \setminus \mathcal{B}$ that is "surrounded" by $\mathcal{T}$. Given these observations, we have the following lemma. The proof of Lemma 7 is presented in Appendix B.6.

**Lemma 7.** *The image* $g_{\ell^*}([0,p_1]) \setminus \mathcal{B}$ *is contained in a bounded path-component of* $\mathbb{R}^2 \setminus \mathcal{T}$ *unless* $g_{\ell^*}([0,p_1]) \cap g_{\ell^*}([p_2,1]) \neq \emptyset$.

Figures 2(b) and 2(c) illustrates the two possible cases of Lemma 7. If $g_{\ell^*}([0,p_1]) \cap g_{\ell^*}([p_2,1]) \neq \emptyset$ (Figure 2(c)), this contradicts (3). Then, $g_{\ell^*}([0,p_1]) \setminus \mathcal{B}$ must be contained in a bounded path-component of $\mathbb{R}^2 \setminus \mathcal{T}$. Recall that $g_{\ell^*}([0,p_1]) \setminus \mathcal{B}$ has to move to $f([0,p_1]) \setminus \mathcal{B}$ by layers $\ell^* + 1, \ldots, L-1$. However, by Lemma 6, if any point in $g_{\ell^*}([0,p_1]) \setminus \mathcal{B}$ moves, then it must intersect with the image of $\mathcal{T}$. If it intersects with the image of $g_{\ell^*}([p_2,1])$, then (3) is violated, hence a contradiction. If it intersects with $\mathcal{B}$ at the $\ell^\dagger$-th layer for some $\ell^\dagger > \ell^*$, then $\mathcal{B} \not\subset \mathcal{S}$ for $\mathcal{S}$ in Lemma 5 determined by $\ell^\dagger$. Hence, it violates the definition of $\ell^*$ as $\phi_{\ell^\dagger}^{-1} \circ \sigma \circ \phi_{\ell^\dagger}(\mathcal{B}) \neq \mathcal{B}$ by Lemma 5. Namely, the approximation by $f$ is impossible in any cases. This completes the proof of Theorem 2.

## 6 CONCLUSION

The universal approximation property of width-bounded networks is one of the fundamental problems in the expressive power theory of deep learning. Prior arts attempt to characterize the minimum width sufficient for universal approximation; however, they only provide upper and lower bounds with large gaps. In this work, we provide the first exact characterization of the minimum width of RELU networks and RELU+STEP networks. In addition, we observe interesting dependence of the minimum width on the target function classes and activation functions, in contrast to the minimum depth of classical results. We believe that our results and analyses would contribute to a better understanding of the performance of modern deep and narrow network architectures.

### ACKNOWLEDGEMENTS

This work was supported by Institute of Information & Communications Technology Planning & Evaluation (IITP) grant funded by the Korea government (MSIT) (No.2019-0-00075, Artificial Intelligence Graduate School Program (KAIST) and No.2017-0-01779, A machine learning and statistical inference framework for explainable artificial intelligence). Chulhee Yun acknowledges financial supports from Korea Foundation for Advanced Studies, NSF CAREER grant 1846088, and ONR grant N00014-20-1-2394.

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

# Appendix

We first provide proofs of upper bounds in Theorems 1, 3, 4 in Appendix A. In Appendix B, we provide proofs of lower bounds in Theorem 1, 3 and proofs of Lemmas 5, 6, 7 used for proving the lower bound in Theorem 2. Throughout Appendix, we denote the coordinate-wise RELU by $\sigma$ and we denote the $i$-th coordinate of an output of a function $f(x)$ by $(f(x))_i$.

## A  PROOFS OF UPPER BOUNDS

### A.1  PROOF OF TIGHT UPPER BOUND IN THEOREM 3

In this section, we prove the tight upper bound on the minimum width in Theorem 3, i.e., width-$(\max\{d_x + 1, d_y\})$ RELU+STEP networks are dense in $C([0,1]^{d_x}, \mathbb{R}^{d_y})$. In particular, we prove that for any $f^* \in C([0,1]^{d_x}, [0,1]^{d_y})$, for any $\varepsilon > 0$, there exists a RELU+STEP network $f$ of width $\max\{d_x + 1, d_y\}$ such that $\sup_{x \in [0,1]^{d_x}} \|f^*(x) - f(x)\|_\infty \le \varepsilon$. Here, we note that the domain and the codomain can be easily generalized to arbitrary compact support and arbitrary codomain, respectively.

Our construction is based on the three-part coding scheme introduced in Section 4.1. First, consider constructing a RELU+STEP network for the encoder. From the definition of $q_K$, one can observe that the mapping is discontinuous and piecewise constant. Hence, the exact construction (or even the uniform approximation) of the encoder requires the use of discontinuous activation functions such as STEP (recall its definition $x \mapsto \mathbf{1}[x \ge 0]$). We introduce the following lemma for the exact construction of $q_K$. The proof of Lemma 8 is presented in Appendix A.4.

**Lemma 8.** *For any $K \in \mathbb{N}$, there exists a RELU+STEP network $f : \mathbb{R} \to \mathbb{R}$ of width 2 and $O(2^K)$ layers such that $f(x) = q_K(x)$ for all $x \in [0,1]$.*

For constructing the encoder via a RELU+STEP network of width $d_x + 1$, we sequentially apply $q_K$ to each input coordinate, by utilizing the extra width 1 and Lemma 8. Here, when we apply $q_K$ to some coordinate (requiring width 2), we preserve other coordinates by applying identity mapping (requiring width $d_x - 1$) for constructing the encoder within width $d_x + 1$. Once we apply $q_K$ for all input coordinates, we apply the linear transformation $\sum_{i=1}^{d_x} q_K(x_i) \times 2^{-(i-1)K}$ to obtain the output of the encoder.

On the other hand, the memorizer only maps a finite number of scalar inputs to the corresponding scalar targets, which can be easily implemented by piecewise linear continuous functions. We show that the memorizer can be exactly constructed by a RELU network of width 2 using the following lemma. The proof of Lemma 9 is presented in Appendix A.5.

**Lemma 9.** *For any function $f^* : \mathbb{R} \to \mathbb{R}$, any finite set $\mathcal{X} \subset \mathbb{R}$, and any compact interval $\mathcal{I} \subset \mathbb{R}$ containing $\mathcal{X}$, there exists a RELU network $f : \mathbb{R} \to \mathbb{R}$ of width 2 and $O(|\mathcal{X}|)$ layers such that $f(x) = f^*(x)$ for all $x \in \mathcal{X}$ and $f(\mathcal{I}) \subset [\min f^*(\mathcal{X}), \max f^*(\mathcal{X})]$.*

Likewise, the decoder maps a finite number of scalar inputs in $\mathcal{C}_{d_y M}$ to corresponding target vectors in $\mathcal{C}_M^{d_y}$. Here, each coordinate of a target vector corresponds to some consequent bits of the binary representation of the input. Under the similar idea used for our implementation of the memorizer, we show that the decoder can be exactly constructed by a RELU network of width $d_y$ using the following lemma. The proof of Lemma 10 is presented in Appendix A.6.

**Lemma 10.** *For any $d_y, M \in \mathbb{N}$, for any $\delta > 0$, there exists a RELU network $f : \mathbb{R} \to \mathbb{R}^2$ of width $d_y$ and $O(2^M)$ layers such that for all $c \in \mathcal{C}_{d_y M}$*

$$f(c) = \texttt{decode}_M(c).$$

*Furthermore, it holds that $f(\mathbb{R}) \subset [0,1]^{d_y}$.*

Finally, as the encoder, the memorizer, and the decoder can be constructed by RELU+STEP networks of width $d_x + 1$, width 2, and width $d_y$, respectively, the width of the overall RELU+STEP network $f$ is $\max\{d_x + 1, d_y\}$. In addition, as mentioned in Section 4.1, choosing $K, M \in \mathbb{N}$ large enough so that $\omega_{f^*}(2^{-K}) + 2^{-M} \le \varepsilon$ ensures $\|f^* - f\|_\infty \le \varepsilon$. This completes the proof of the tight upper bound in Theorem 3.

## A.2 PROOF OF TIGHT UPPER BOUND IN THEOREM 1

In this section, we derive the upper bound in Theorem 1. In particular, we prove that for any $p \in [1, \infty)$, for any $f^* \in L^p(\mathbb{R}^{d_x}, \mathbb{R}^{d_y})$, for any $\varepsilon > 0$, there exists a RELU network $f$ of width $\max\{d_x + 1, d_y\}$ such that $\|f^* - f\|_p \leq \varepsilon$. To this end, we first note that since $f^* \in L^p(\mathbb{R}^{d_x}, \mathbb{R}^{d_y})$, there exists a continuous function $f'$ on a compact support such that

$$\|f^* - f'\|_p \leq \frac{\varepsilon}{2}.$$

Namely, if we construct a RELU network $f$ such that $\|f' - f\|_p \leq \frac{\varepsilon}{2}$, then it completes the proof. Throughout this proof, we assume that the support of $f'$ is a subset of $[0,1]^{d_x}$ and its codomain to be $[0,1]^{d_y}$ which can be easily generalized to arbitrary compact support and arbitrary codomain, respectively.

We approximate $f'$ by a RELU network using the three-part coding scheme introduced in Section 4.1. We will refer to our implementations of the three parts as $\texttt{encode}_K^\dagger(x)$, $\texttt{memorize}_{K,M}^\dagger$, and $\texttt{decode}_M^\dagger$. That is, we will approximate $f'$ by a RELU network

$$f := \texttt{decode}_M^\dagger \circ \texttt{memorize}_{K,M}^\dagger \circ \texttt{encode}_K^\dagger.$$

However, unlike our construction of RELU+STEP networks in Appendix A.1, STEP is not available, i.e., uniform approximation of $q_K$ is impossible. Nevertheless, one can approximate $q_K$ with some continuous piecewise linear function by approximating regions around discontinuities with some linear functions. Under this idea, we introduce the following lemma. The proof of Lemma 11 is presented in Appendix A.7.

**Lemma 11.** *For any $d_x, K \in \mathbb{N}$, for any $\gamma > 0$, there exist a RELU network $f : \mathbb{R}^{d_x} \to \mathbb{R}$ of width $d_x + 1$ and $O(2^K)$ layers and $\mathcal{D}_\gamma \subset [0,1]^{d_x}$ such that for all $x \in [0,1]^{d_x} \setminus \mathcal{D}_\gamma$,*

$$f(x) = \texttt{encode}_K(x),$$

*$\mu(\mathcal{D}_\gamma) < \gamma$, $f(\mathcal{D}_\gamma) \subset [0,1]$, and $f(\mathbb{R}^{d_x} \setminus [0,1]^{d_x}) = \{1 - 2^{d_x K}\}$ where $\mu$ denotes the Lebesgue measure.*

By Lemma 11, there exist a RELU network $\texttt{encode}_K^\dagger$ of width $d_x + 1$ and $\mathcal{D}_\gamma \subset [0,1]^{d_x}$ such that $\mu(\mathcal{D}_\gamma) < \gamma$ and

$$\begin{aligned}
\texttt{encode}_K^\dagger(x) &= \texttt{encode}_K(x) \quad \text{for all } x \in [0,1]^{d_x} \setminus \mathcal{D}_\gamma, \\
\texttt{encode}_K^\dagger(\mathbb{R}^{d_x} \setminus [0,1]^{d_x}) &= \{1 - 2^{d_x K}\}.
\end{aligned} \tag{4}$$

We approximate the encoder by $\texttt{encode}_K^\dagger$. Here, we note that inputs from $\mathcal{D}_\gamma$ would be mapped to arbitrary values by $\texttt{encode}_K^\dagger$. Nevertheless, it is not critical to the error $\|f' - f\|_p$ as $\mu(\mathcal{D}_\gamma) < \gamma$ can be made arbitrarily small by choosing a sufficiently small $\gamma$.

The implementation of the memorizer utilizes Lemma 9 as in Appendix A.1. However, as $f'(x) = 0$ for all $x \in \mathbb{R}^{d_x} \setminus [0,1]^{d_x}$, we construct a RELU network $\texttt{memorize}_{K,M}^\dagger$ of width 2 so that

$$\texttt{memorize}_{K,M}^\dagger \left( \texttt{encode}_{K,L}^\dagger(\mathbb{R}^{d_x} \setminus [0,1]^{d_x}) \right) = \{0\}.$$

To achieve this, we design the memorizer for $c \in \mathcal{C}_{d_x K}$ using Lemma 9 and based on (4) as

$$\texttt{memorize}_{K,M}^\dagger(c) = \begin{cases} 0 & \text{if } c = 1 - 2^{d_x K} \\ \texttt{memorize}_{K,M}(c) & \text{otherwise} \end{cases}.$$

We note that such a design incurs an undesired error that a subset of $\mathcal{E}_K := [1 - 2^{-K}, 1]^{d_x}$ might be mapped to zero after applying $\texttt{memorize}_{K,M}^\dagger$. Nevertheless, mapping $\mathcal{E}_K$ to zero is not critical to the error $\|f' - f\|_p$ as $\mu(\mathcal{E}_K) < 2^{-d_x K}$ can be made arbitrarily small by choosing a sufficiently large $K$.

We implement the decoder by a RELU network $\texttt{decode}_M^\dagger$ of width $d_y$ using Lemma 10 as in Appendix A.1. Then, by Lemma 10, it holds that $\texttt{decode}_M^\dagger(\mathbb{R}) \subset [0,1]^{d_y}$, and hence, $f(\mathbb{R}^{d_x}) \subset [0,1]^{d_y}$.

Finally, we bound the error $\|f' - f\|_p$ utilizing the following inequality:

$$\|f' - f\|_p = \left( \int_{\mathbb{R}^{d_x}} \|f'(x) - f(x)\|_p^p dx \right)^{\frac{1}{p}}$$

$$= \left( \int_{[0,1]^{d_x} \setminus (\mathcal{E}_K \cup \mathcal{D}_\gamma)} \|f'(x) - f(x)\|_p^p dx + \int_{\mathcal{E}_K \cup \mathcal{D}_\gamma} \|f'(x) - f(x)\|_p^p dx \right)^{\frac{1}{p}}$$

$$\leq \left( d_y (\omega_{f'}(2^{-K}) + 2^{-M})^p + (\mu(\mathcal{E}_K) + \mu(\mathcal{D}_\gamma)) \times \sup_{x \in \mathcal{E}_K \cup \mathcal{D}_\gamma} \|f'(x) - f(x)\|_p^p \right)^{\frac{1}{p}}$$

$$< \left( d_y (\omega_{f'}(2^{-K}) + 2^{-M})^p + (2^{-d_x K} + \gamma) \times \left( \sup_{x \in [0,1]^{d_x}} \|f'(x)\|_p + \sup_{x \in [0,1]^{d_x}} \|f(x)\|_p \right)^p \right)^{\frac{1}{p}}$$

$$\leq \left( d_y (\omega_{f'}(2^{-K}) + 2^{-M})^p + (2^{-d_x K} + \gamma) \times \left( \sup_{x \in [0,1]^{d_x}} \|f'(x)\|_p + (d_y)^{\frac{1}{p}} \right)^p \right)^{\frac{1}{p}}.$$

By choosing sufficiently large $K, M$ and sufficiently small $\gamma$, one can make the RHS smaller than $\varepsilon/2$ as $\sup_{x \in [0,1]^{d_x}} \|f'(x)\|_p < \infty$. This completes the proof of the tight upper bound in Theorem 1.

## A.3    PROOF OF THEOREM 4

In this section, we prove Theorem 4 by proving the following statement: For any $p \in [1, \infty)$, for any $f^* \in L^p(\mathcal{K}, \mathbb{R}^{d_y})$, for any $\varepsilon > 0$, there exists a $\rho$ network $f$ of width $\max\{d_x + 2, d_y + 1\}$ such that $\|f^* - f\|_p \leq \varepsilon$. Here, there exists a continuous function $f' \in C(\mathcal{K}, \mathbb{R}^{d_y})$ such that

$$\|f^* - f'\|_p \leq \frac{\varepsilon}{2}$$

since $f^* \in L^p(\mathcal{K}, \mathbb{R}^{d_y})$. Namely, if we construct a $\rho$ network $f$ such that $\|f' - f\|_p \leq \frac{\varepsilon}{2}$, it completes the proof. Throughout the proof, we assume that the support of $f'$ is a subset of $[0,1]^{d_x}$ and its codomain is $[0,1]^{d_y}$ which can be easily generalized to arbitrary compact support and arbitrary codomain, respectively.

Before describing our construction, we first introduce the following lemma.

**Lemma 12** [Kidger and Lyons (2020, Proposition 4.9)]*. Let $\rho : \mathbb{R} \to \mathbb{R}$ be any continuous non-polynomial function which is continuously differentiable at some $z$ with $\rho'(z) \neq 0$. Then, for any $f^* \in C(\mathcal{K}, \mathbb{R}^{d_y})$, for any $\varepsilon > 0$, there exists a $\rho$ network $f : \mathcal{K} \to \mathbb{R}^{d_x} \times \mathbb{R}^{d_y}$ of width $d_x + d_y + 1$ such that for all $x \in \mathcal{K}$,*

$$f(x) := (y_1(x), y_2(x)), \text{ where } \|y_1(x) - x\|_\infty \leq \varepsilon \quad \text{and} \quad \|y_2(x) - f^*(x)\|_\infty \leq \varepsilon.$$

We note that Proposition 4.9 by Kidger and Lyons (2020) only ensures $\|y_2(x) - f^*(x)\|_\infty \leq \varepsilon$; however, its proof provides $\|y_1(x) - x\|_\infty \leq \varepsilon$ as well.

The proof of Theorem 4 also utilizes our coding scheme; here, we approximate RELU network constructions $\mathtt{encode}_K^\dagger$, $\mathtt{memorize}_{K,M}^\dagger$, and $\mathtt{decode}_M^\dagger$ in Appendix A.2 by $\rho$ networks. Using Lemma 12, for any $\varepsilon_1 > 0$, we approximate $\mathtt{encode}_K^\dagger$ by a $\rho$ network $\mathtt{encode}_K^\ddagger$ of width $d_x + 2$ so that

$$\left\| \mathtt{encode}_K^\ddagger(x) - \mathtt{encode}_K^\dagger(x) \right\|_\infty \leq \varepsilon_1 \quad \text{for all} \quad x \in [0,1]^{d_x} \setminus \mathcal{D}_\gamma$$

and $\mathtt{encode}_K^\ddagger([0,1]^{d_x}) \subset [-\varepsilon_1, 1+\varepsilon_1]$. We note that $\mathtt{encode}_K^\ddagger([0,1]^{d_x}) \subset [-\varepsilon_1, 1+\varepsilon_1]$ is possible as $\mathtt{encode}_K^\dagger(\mathbb{R}^{d_x}) \subset [0,1]$ by Lemma 11.

Approximating the memorizer can be done in a similar manner. Using Lemma 12, for any compact interval $\mathcal{I}_2 \subset \mathbb{R}$ containing $\mathcal{C}_{d_x K}$, for any $\varepsilon_2 > 0$, we approximate $\mathtt{memorize}_{K,M}^\dagger$ by a $\rho$ network $\mathtt{memorize}_{K,M}^\ddagger$ of width 3 so that

$$\left\| \mathtt{memorize}_{K,M}^\ddagger(c) - \mathtt{memorize}_{K,M}^\dagger(c) \right\|_\infty \leq \varepsilon_2 \quad \text{for all} \quad c \in \mathcal{C}_{d_x K}$$

and $\texttt{memorize}^{\ddagger}_{K,M}(\mathcal{I}_2) \subset [-\varepsilon_2, 1 + \varepsilon_2]$. We note that $\texttt{memorize}^{\ddagger}_{K,M}(\mathcal{I}_2) \subset [-\varepsilon_2, 1 + \varepsilon_2]$ is possible as there exists $\texttt{memorize}^{\dagger}_{K,M}$ (i.e., a RELU network) such that $\texttt{memorize}^{\dagger}_{K,M}(\mathcal{I}_2) \subset [0, 1]$ by Lemma 9.

For approximating the decoder, we introduce the following lemma. The proof of Lemma 13 is presented in Appendix A.8.

**Lemma 13.** *For any $d_y, M \in \mathbb{N}$, for any $\varepsilon > 0$, for any compact interval $\mathcal{I} \subset \mathbb{R}$ containing $[0,1]$, there exists a $\rho$ network $f : \mathbb{R} \to \mathbb{R}^{d_y}$ of width $d_y + 1$ such that for all $c \in \mathcal{I}$,*

$$\|f(c) - \texttt{decode}^{\dagger}_M(c)\|_\infty \leq \varepsilon.$$

*Namely, $f(\mathcal{I}) \subset [-\varepsilon, 1 + \varepsilon]^{d_y}$.*

By Lemma 13, for any compact interval $\mathcal{I}_3 \subset \mathbb{R}$ containing $[0,1]$, for any $\varepsilon_3 > 0$, there exists a $\rho$ network $\texttt{decode}^{\ddagger}_M$ of width $d_y + 1$ such that

$$\left\| \texttt{decode}^{\ddagger}_M(c) - \texttt{decode}^{\dagger}_M(c) \right\|_\infty \leq \varepsilon_3 \quad \text{for all} \quad c \in \mathcal{C}_{d_y M}$$

and $\texttt{decode}^{\ddagger}_M(\mathcal{I}_3) \in [-\varepsilon_3, 1 + \varepsilon_3]^{d_y}$.

We approximate $f'$ by a $\rho$ network $f$ of width $\max\{d_x + 2, d_y + 1\}$ defined as

$$f := \texttt{decode}^{\ddagger}_M \circ \texttt{memorize}^{\ddagger}_{K,M} \circ \texttt{encode}^{\ddagger}_K.$$

Here, for any $\eta > 0$, by choosing sufficiently large $K, M$, sufficiently large $\mathcal{I}_2, \mathcal{I}_3$, and sufficiently small $\varepsilon_1, \varepsilon_2, \varepsilon_3$ so that $\omega_{f'}(2^{-K}) + 2^{-M} \leq \frac{\eta}{2}$ and $\omega_{\texttt{decode}^{\ddagger}_M}\left(\omega_{\texttt{memorize}^{\ddagger}_{K,M}}(\varepsilon_1) + \varepsilon_2\right) + \varepsilon_3 \leq \frac{\eta}{2}$, we have

$$\sup_{x \in [0,1]^{d_x} \setminus \mathcal{D}_\gamma} \|f'(x) - f(x)\|_\infty \leq \eta \quad \text{and} \quad f([0,1]^{d_x}) \subset [-\tfrac{\eta}{2}, 1 + \tfrac{\eta}{2}]^{d_x} \tag{5}$$

where $\omega_{\texttt{memorize}^{\ddagger}_{K,M}}$ and $\omega_{\texttt{decode}^{\ddagger}_M}$ are defined on $\mathcal{I}_2$ and $\mathcal{I}_3$, respectively.

Finally, we bound the error $\|f' - f\|_p$ utilizing the following inequality:

$$\|f' - f\|_p = \left( \int_{[0,1]^{d_x}} \|f'(x) - f(x)\|_p^p dx \right)^{\frac{1}{p}}$$

$$= \left( \int_{[0,1]^{d_x} \setminus \mathcal{D}_\gamma} \|f'(x) - f(x)\|_p^p dx + \int_{\mathcal{D}_\gamma} \|f'(x) - f(x)\|_p^p dx \right)^{\frac{1}{p}}$$

$$\leq \left( \sup_{x \in [0,1]^{d_x} \setminus \mathcal{D}_\gamma} \|f'(x) - f(x)\|_p^p + \mu(\mathcal{D}_\gamma) \times \sup_{x \in \mathcal{D}_\gamma} \|f'(x) - f(x)\|_p^p \right)^{\frac{1}{p}}$$

$$\leq \left( \sup_{x \in [0,1]^{d_x} \setminus \mathcal{D}_\gamma} \|f'(x) - f(x)\|_p^p + \gamma \times \left( \sup_{x \in [0,1]^{d_x}} \|f'(x)\|_p + \sup_{x \in [0,1]^{d_x}} \|f(x)\|_p \right)^p \right)^{\frac{1}{p}}.$$

By choosing sufficiently small $\varepsilon_1, \varepsilon_2, \varepsilon_3, \gamma$, sufficiently large $K, M$, and sufficiently large $\mathcal{I}_2, \mathcal{I}_3$, one can make the RHS smaller than $\varepsilon/2$ due to (5) and the fact that $\sup_{x \in [0,1]^{d_x}} \|f'(x)\|_p < \infty$. This completes the proof of Theorem 4.

### A.4 PROOF OF LEMMA 8

We construct $f : \mathbb{R} \to \mathbb{R}$ as $f(x) := f_{2^K} \circ \cdots \circ f_1(x)$ where each $f_\ell : \mathbb{R} \to \mathbb{R}$ is defined for $x \in [0,1]$ as

$$f_\ell(x) := \begin{cases} (\ell - 1) \times 2^{-K} & \text{if } x \in [(\ell - 1) \times 2^{-K}, \ell \times 2^{-K}) \\ x & \text{if } x \notin [(\ell - 1) \times 2^{-K}, \ell \times 2^{-K}) \end{cases}$$

$$= g_{\ell 3} \circ g_{\ell 2} \circ g_{\ell 1}(x)$$

where $g_{\ell 1} : \mathbb{R} \to \mathbb{R}^2$, $g_{\ell 2} : \mathbb{R}^2 \to \mathbb{R}^2$, and $g_{\ell 3} : \mathbb{R}^2 \to \mathbb{R}$ are defined as

$$g_{\ell 1}(x) := \big(\sigma(x), \sigma(x - \ell)\big)$$
$$g_{\ell 2}(x, z) := \big(\sigma(x + z), -\sigma(x - \ell + 1)\big)$$
$$g_{\ell 3}(x, z) := \sigma(x + z) + \mathbf{1}[x \geq \ell].$$

This directly implies that $f(x) = q_K(x)$ for all $x \in [0, 1]$ and completes the proof of Lemma 8.

## A.5   PROOF OF LEMMA 9

Let $|\mathcal{X}| = N$ and $x^{(1)}, \ldots, x^{(N)}$ be distinct elements of $\mathcal{X}$ in an increasing order, i.e., $x^{(i)} < x^{(j)}$ if $i < j$. Let $x^{(0)} := \min \mathcal{I}$ and $x^{(N+1)} := \max \mathcal{I}$. Here, $x^{(0)} \leq x^{(1)}$ and $x^{(N)} \leq x^{(N+1)}$ as $\mathcal{X} \subset \mathcal{I}$. Without loss of generality, we assume that $x^{(0)} = 0$. Consider a continuous piecewise linear function $f^\dagger : [x^{(0)}, x^{(N+1)}] \to \mathbb{R}$ of $N + 1$ linear pieces defined as

$$f^\dagger(x) := \begin{cases} f^*(x^{(1)}) & \text{if } x \in [x^{(0)}, x^{(1)}) \\ f^*(x^{(i)}) + \frac{f^*(x^{(i+1)}) - f^*(x^{(i)})}{x^{(i+1)} - x^{(i)}}(x - x^{(i)}) & \text{if } x \in [x^{(i)}, x^{(i+1)}) \text{ for } 1 \leq i \leq N - 1 \ . \\ f^*(x^{(N)}) & \text{if } x \in [x^{(N)}, x^{(N+1)}] \end{cases}$$

Now, we introduce the following lemma.

**Lemma 14.** *For any compact interval $\mathcal{I} \subset \mathbb{R}$, for any continuous piecewise linear function $f^* : \mathcal{I} \to \mathbb{R}$ with $P$ linear pieces, there exists a RELU network $f$ of width 2 and $O(P)$ layers such that $f^*(x) = f(x)$ for all $x \in \mathcal{I}$.*

From Lemma 14, there exists a RELU network $f$ of width 2 and $O(|\mathcal{X}|)$ layers such that $f^\dagger(x) = f(x)$ for all $x \in \mathcal{X}$. Since $\mathcal{X} \subset [x^{(0)}, x^{(N+1)}] = \mathcal{I}$ and $f^\dagger(\mathcal{I}) \subset \big[\min f^*(\mathcal{X}), \max f^*(\mathcal{X})\big]$, this completes the proof of Lemma 9.

*Proof of Lemma 14.* Suppose that $f^*$ is linear on intervals $[\min \mathcal{I}, x_1), [x_1, x_2), \ldots, [x_{P-1}, \max \mathcal{I}]$ and parametrized as

$$f^*(x) = \begin{cases} a_1 \times x + b_1 & \text{if } x \in [\min \mathcal{I}, x_1) \\ a_2 \times x + b_2 & \text{if } x \in [x_1, x_2) \\ \quad \vdots & \\ a_P \times x + b_P & \text{if } x \in [x_{P-1}, \max \mathcal{I}] \end{cases}$$

for some $a_i, b_i \in \mathbb{R}$ satisfying $a_i \times x_i + b_i = a_{i+1} \times x_i + b_{i+1}$. Without loss of generality, we assume that $\min \mathcal{I} = 0$.

Now, we prove that for any $P \geq 1$, there exists a RELU network $f : \mathcal{I} \to \mathbb{R}^2$ of width 2 and $O(P)$ layers such that $(f(x))_1 = \sigma(x - x_{P-1})$ and $(f(x))_2 = f^*(x)$. Then, $(f(x))_2$ is the desired RELU network and completes the proof. We use the mathematical induction on $P$ for proving the existence of such $f$. If $P = 1$, choosing $(f(x))_1 = \sigma(x)$ and $(f(x))_2 = a_1 \times \sigma(x) + b_1$ completes the construction of $f$. Now, consider $P > 1$. From the induction hypothesis, there exists a RELU network $g$ of width 2 such that

$$(g(x))_1 = \sigma(x - x_{P-2})$$

$$(g(x))_2 = \begin{cases} a_1 \times x + b_1 & \text{if } x \in [\min \mathcal{I}, x_1) \\ a_2 \times x + b_2 & \text{if } x \in [x_1, x_2) \\ \quad \vdots & \\ a_{P-1} \times x + b_{P-1} & \text{if } x \in [x_{P-2}, \max \mathcal{I}] \end{cases}.$$

Then, the following construction of $f$ completes the proof of the mathematical induction:

$$f(x) = h_2 \circ h_1 \circ g(x)$$
$$h_1(x, z) = \big(\sigma(x - x_{P-1} + x_{P-2}), \sigma(z - K) + K\big)$$
$$h_2(x, z) = \big(x, z + (a_{P-1} - a_{P-2}) \times x\big)$$

where $K := \min_i \min_{x \in \mathcal{I}}\{a_i \times x + b_i\}$. This completes the proof of Lemma 14. $\quad\square$

### A.6 PROOF OF LEMMA 10

Before describing our proof, we first introduce the following lemma. The proof of Lemma 15 is presented in Appendix A.9.

**Lemma 15.** *For any $M \in \mathbb{N}$, for any $\delta > 0$, there exists a* RELU *network $f : \mathbb{R} \to \mathbb{R}^2$ of width 2 and $O(2^M)$ layers such that for all $x \in [0, 1] \setminus \mathcal{D}_{M,\delta}$,*

$$f(x) := (y_1(x), y_2(x)), \quad where \quad y_1(x) = q_M(x), \quad y_2(x) = 2^M \times (x - q_M(x)), \quad (6)$$

*and $\mathcal{D}_{M,\delta} := \bigcup_{i=1}^{2^M-1}(i \times 2^{-M} - \delta, i \times 2^{-M})$. Furthermore, it holds that*

$$f(\mathbb{R}) \subset [0, 1 - 2^{-M}] \times [0, 1]. \quad (7)$$

In Lemma 15, one can observe that $\mathcal{C}_{d_y M} \subset [0, 1] \setminus \mathcal{D}_{M,\delta}$ for $\delta < 2^{-d_y M}$, i.e., there exists a RELU network $g$ of width 2 satisfying (6) on $\mathcal{C}_{d_y M}$ and (7). $g$ enables us to extract the first $M$ bits of the binary representation of $c \in \mathcal{C}_{d_y M}$. Consider outputs of $g(c)$: $(g(c))_1$ for $c \in \mathcal{C}_{d_y M}$ is the first coordinate of $\text{decode}_M(c)$ while $(g(c))_2 \in \mathcal{C}_{(d_y-1)M}$ contains information on other coordinates of $\text{decode}_M(c)$. Now, consider further applying $g$ to $(g(c))_2$ and passing though the output $(g(c))_1$ via the identity function (RELU is identity for positive inputs). Then, $\left(g\left((g(c))_2\right)\right)_1$ is the second coordinate of $\text{decode}_M(c)$ while $\left(g\left((g(c))_2\right)\right)_2$ contains information on coordinates other than the first and the second ones of $\text{decode}_M(c)$. Under this observation, if we iteratively apply $g$ to the second output of the prior $g$ and pass through all first outputs of previous $g$'s, then we recover all coordinates of $\text{decode}_M(c)$ within $d_y - 1$ applications of $g$. Note that both the first and the second outputs of the $(d_y - 1)$-th $g$ correspond to the second last and the last coordinate of $\text{decode}_M(c)$, respectively. Our construction of $f$ is such an iterative $d_y - 1$ applications of $g$ which can be implemented by a RELU network of width $d_y$. Here, (7) in Lemma 15 enables us to achieve $f(\mathbb{R}) \subset [0, 1]^{d_y}$. This completes the proof of Lemma 10.

### A.7 PROOF OF LEMMA 11

To begin with, we introduce the following Lemma. The proof of Lemma 16 is presented in Appendix A.10.

**Lemma 16.** *For any $d_x$, for any $\alpha \in (0, 0.5)$, there exists a* RELU *network $f : \mathbb{R}^{d_x} \to \mathbb{R}^{d_x}$ of width $d_x + 1$ and $O(1)$ layers such that $f(x) = (1, \ldots, 1)$ for all $x \in \mathbb{R}^{d_x} \setminus [0, 1]^{d_x}$, $f(x) = x$ for all $x \in [\alpha, 1 - \alpha]^{d_x}$, and $f(\mathbb{R}^{d_x}) \subset [0, 1]^{d_x}$.*

By Lemma 16, there exists a RELU network $h_1$ of width $d_x + 1$ and $O(1)$ layers such that $h_1(x) = (1, \ldots, 1)$ for all $x \in \mathbb{R}^{d_x} \setminus [0, 1]^{d_x}$, $h_1(x) = x$ for all $x \in [\alpha, 1 - \alpha]^{d_x}$, and $h_1(\mathbb{R}^{d_x}) \subset [0, 1]^{d_x}$. Furthermore, by Lemma 15, for any $\delta > 0$, there exists a RELU network $g : \mathbb{R} \to \mathbb{R}$ of width 2 and $O(2^K)$ layers such that $g(c) = q_K(c)$ for all $c \in [0, 1] \setminus \mathcal{D}_{K,\delta}$ (see Lemma 15 for the definition of $\mathcal{D}_{K,\delta}$).

We construct a network $h_2 : \mathbb{R}^{d_x} \to \mathbb{R}^{d_x}$ of width $d_x + 1$ by sequentially applying $g$ for each coordinate of an input $x \in \mathbb{R}^{d_x}$, utilizing the extra width 1. Then, $h_2(x) = q_K(x)$ for all $x \in [0, 1]^{d_x} \setminus \mathcal{D}_{K,\delta,d_x}$ where

$$\mathcal{D}_{K,\delta,d_x} := \{x \in \mathbb{R}^{d_x} : x_i \in \mathcal{D}_{K,\delta} \text{ for some } i\}.$$

Note that we use $q_K(x)$ for denoting the coordinate-wise $q_K$ for a vector $x$.

Now, we define $\mathcal{D}_\gamma := ([0, 1]^{d_x} \setminus [\alpha, 1 - \alpha]^{d_x}) \cup \mathcal{D}_{K,\delta,d_x} \subset [0, 1]^{d_x}$. Then, from constructions of $h_1$ and $h_2$, we have

$$
\begin{aligned}
h_2 \circ h_1(x) &= q_K(x) && \text{for all } x \in [0, 1]^{d_x} \setminus \mathcal{D}_\gamma \\
h_2 \circ h_1(x) &= (1 - 2^{-K}, \ldots, 1 - 2^{-K}) && \text{for all } x \in \mathbb{R}^{d_x} \setminus [0, 1]^{d_x} \\
h_2 \circ h_1(x) &\subset [0, 1 - 2^{-K}]^{d_x} && \text{for all } x \in \mathcal{D}_\gamma
\end{aligned}
$$

where we use the fact that $(1, \ldots, 1) \notin \mathcal{D}_{K,\delta,d_x}$ and $q_K((1, \ldots, 1)) = (1 - 2^{-K}, \ldots, 1 - 2^{-K})$.

Finally, we construct a RELU network $f$ of width $d_x + 1$ as

$$f(x) := \sum_{i=1}^{d_x} (h_2 \circ h_1(x))_i \times 2^{-(i-1)K}.$$

Then, it holds that

$$
\begin{aligned}
f(x) &= \mathtt{encode}_K(x) && \text{for all } x \in [0,1]^{d_x} \setminus \mathcal{D}_\gamma \\
f(x) &= 1 - 2^{d_x K} && \text{for all } x \in \mathbb{R}^{d_x} \setminus [0,1]^{d_x} \\
f(x) &\subset [0,1] && \text{for all } x \in \mathcal{D}_\gamma.
\end{aligned}
$$

In addition, if we choose sufficiently small $\alpha$ and $\delta$ so that $\mu(\mathcal{D}_\gamma) < \gamma$, then $f$ satisfies all conditions in Lemma 11. This completes the proof of Lemma 11.

### A.8 PROOF OF LEMMA 13

The proof of Lemma 13 is almost identical to that of Lemma 10. In particular, we approximate the RELU network construction of iterative $d_y - 1$ applications of $g$ (see Appendix A.6 for the definition of $g$) by a $\rho$ network of width $d_y + 1$. To this end, we consider a $\rho$ network $h$ of width 3 approximating $g$ on some interval $\mathcal{J}$ within $\alpha$ error using Lemma 12. Then, one can observe that iterative $d_y - 1$ applications of $h$ (as in Appendix A.6) results in a $\rho$ network $f$ of width $d_y + 1$. Here, passing through the identity function can be approximated using a $\rho$ network of width 1, i.e., same width to RELU networks (see Lemma 4.1 by Kidger and Lyons (2020) for details). Furthermore, since $h$ is uniformly continuous on $\mathcal{J}$, it holds that $\|f(c) - \mathtt{decode}_M(c)\|_\infty \le \varepsilon$ for all $c \in \mathcal{C}_{d_y M}$ and $f(\mathcal{I}) \subset [-\varepsilon, 1+\varepsilon]^{d_y}$ by choosing sufficiently large $\mathcal{J}$ and sufficiently small $\alpha$ so that $\omega_h(\cdots \omega_h(\omega_h(\alpha) + \alpha) \cdots) + \alpha \le \varepsilon$.[8] This completes the proof of Lemma 13.

### A.9 PROOF OF LEMMA 15

We first clip the input to be in $[0,1]$ using the following RELU network of width 1.

$$\min\{\max\{x, 0\}, 1\} = 1 - \sigma(1 - \sigma(x))$$

After that, we apply $g_\ell : [0,1] \to [0,1]^2$ defined as

$$(g_\ell(x))_1 := x$$

$$(g_\ell(x))_2 := \begin{cases} 0 & \text{if } x \in [0, 2^{-M} - \delta] \\ \delta^{-1} 2^{-M} \times (x - 2^{-M} + \delta) & \text{if } x \in (2^{-M} - \delta, 2^{-M}) \\ 2^{-M} & \text{if } x \in [2^{-M}, 2 \times 2^{-M} - \delta] \\ \delta^{-1} 2^{-M} \times (x - 2 \times 2^{-M} + \delta) + 2^{-M} & \text{if } x \in (2 \times 2^{-M} - \delta, 2 \times 2^{-M}) \\ \vdots \\ (\ell - 1) \times 2^{-M} & \text{if } x \in [(\ell - 1) \times 2^{-M}, 1] \end{cases} \quad (8)$$

From the above definition of $g_\ell$, one can observe that $(g_{2^M}(x))_2 = q_M(x)$ for $x \in [0,1] \setminus \mathcal{D}_{K,\delta}$. Once we implement a RELU network $g$ of width 2 such that $g(x) = g_{2^M}(x)$, then, constructing $f$ as

$$f(x) := \big((g(z))_2, 2^M \times ((g(z))_1 - (g(z))_2)\big)$$

$$z := \min\{\max\{x, 0\}, 1\}$$

completes the proof. Note that as $(g(x))_2 \le x = (g(x))_1$ for all $x \in [0,1]$, $f(x) \in [0, 1 - 2^{-M}] \times [0,1]$. Now, we describe how to construct $g_{2^M}$ by a RELU network. One can observe that $(g_1(x))_2 = 0$ and

$$(g_{\ell+1}(x))_2 = \min\left\{\ell \times 2^{-M}, \max\left\{\delta^{-1} 2^{-M} \times (x - \ell \times 2^{-M} + \delta) + (\ell - 1) \times 2^{-M}, g_\ell(x)\right\}\right\}$$

for all $x$, i.e., alternating applications of $\min\{\cdot, \cdot\}$ and $\max\{\cdot, \cdot\}$. Finally, we introduce the following definition and lemma.

---

[8] We consider $\omega_h$ on $\mathcal{J}$.

**Definition 1** [Hanin and Sellke (2017)]. $f : \mathbb{R}^{d_x} \to \mathbb{R}^{d_y}$ *is a max-min string of length L if there exist affine transformations* $h_1, \ldots, h_L$ *such that*

$$h(x) = \tau_{L-1}(h_L(x), \tau_{L-2}(h_{L-1}(x), \cdots, \tau_2(h_3(x), \tau_1(h_2(x), h_1(x))), \cdots)$$

*where each* $\tau_\ell$ *is either a coordinate-wise* $\max\{\cdot, \cdot\}$ *or* $\min\{\cdot, \cdot\}$.

**Lemma 17** [Hanin and Sellke (2017, Proposition 2)]. *For any max-min string* $f^* : \mathbb{R}^{d_x} \to \mathbb{R}^{d_y}$ *of length L, for any compact* $\mathcal{K} \subset \mathbb{R}^{d_x}$, *there exists a* RELU *network* $f : \mathbb{R}^{d_x} \to \mathbb{R}^{d_x} \times \mathbb{R}^{d_y}$ *of L layers and width* $d_x + d_y$ *such that for all* $x \in \mathcal{K}$,

$$f(x) = (y_1(x), y_2(x)), \quad \text{where} \quad y_1(x) = x \quad \text{and} \quad y_2(x) = f^*(x).$$

We note that Proposition 2 by Hanin and Sellke (2017) itself only ensures $y_2 = f^*(x)$; however, its proof provides $y_1 = x$ as well.

From the definition of the max-min string, one can observe that $(g_{2^M}(x))_2$ is a max-min string. Hence, by Lemma 17, there exists a RELU network $g$ of width 2 such that $g(x) = g_{2^M}(x) = q_M(x)$ for all $x \in \mathcal{D}_{K,\delta}$. This completes the proof of Lemma 15.

### A.10 PROOF OF LEMMA 16

Consider the following two functions from $\mathbb{R}$ to $\mathbb{R}$:

$$h_1(x) := \begin{cases} 0 & \text{if } x \leq 1 - \alpha \\ \frac{1}{\alpha}(x - 1 + \alpha) & \text{if } x \in (1 - \alpha, 1) \\ 1 & \text{if } x \geq 1 \end{cases}$$
$$= \sigma(1 - \sigma(1 - x)/\alpha)$$

$$h_2(x) := \begin{cases} 1 & \text{if } x \leq 0 \\ \frac{1}{\alpha}(\alpha - x) & \text{if } x \in (0, \alpha) \\ 0 & \text{if } x \geq \alpha \end{cases}$$
$$= 1 - \sigma(1 - \sigma(\alpha - x)/\alpha). \tag{9}$$

Using $h_1$ and $h_2$, we first map all $x \in \mathbb{R}^{d_x} \setminus [0, 1]^{d_x}$ to some vector whose coordinates are greater than one via $g : \mathbb{R}^{d_x} \to \mathbb{R}^{d_x}$, defined as

$$g := r_{d_x} \circ s_{d_x} \cdots \circ r_1 \circ s_1$$
$$r_\ell(x) := \sigma(x + 1) - 1 + 10 \times h_1(x_\ell)$$
$$s_\ell(x) := \sigma(x + 1) - 1 + 10 \times h_2(x_\ell).$$

Here we use the addition between a vector and a scalar for denoting addition of the scalar to each coordinate of the vector. Then, one can observe that if $x \in [\alpha, 1 - \alpha]^{d_x}$, then $g(x) = x$ and if $x \in \mathbb{R}^{d_x} \setminus [0, 1]^{d_x}$, then each coordinate of $g(x)$ is greater than one. Furthermore, each $r_\ell$ (or $s_\ell$) can be implemented by a RELU network of width $d_x + 1$ (width $d_x$ for computing $\sigma(x + 1) - 1$ and width one for computing $h_1(x_\ell)$) due to (9). Hence, $g$ can be implemented by a RELU network of width $d_x + 1$.

Finally, we construct a RELU network $f : \mathbb{R}^{d_x} \to \mathbb{R}^{d_x}$ of width $d_x + 1$ as

$$f(x) := \min \{ \max\{g(x), 0\}, 1\}$$
$$\min \{ \max\{x, 0\}, 1\} = 1 - \sigma(1 - \sigma(x)).$$

Then, one can observe that if $x \in [\alpha, 1 - \alpha]^{d_x}$, then $f(x) = x$ and if $x \in \mathbb{R}^{d_x} \setminus [0, 1]^{d_x}$, then $f(x) = (1, \ldots, 1)$, and $f(\mathbb{R}^{d_x}) \subset [0, 1]^{d_x}$. This completes the proof of Lemma 16.

# B  PROOFS OF LOWER BOUNDS

## B.1  PROOF OF GENERAL LOWER BOUND

In this section, we prove that neural networks of width $d_y - 1$ is not dense in both $L^p(\mathcal{K}, \mathbb{R}^{d_y})$ and $C(\mathcal{K}, \mathbb{R}^{d_y})$, regardless of the activation functions.

**Lemma 18.** *For any set of activation functions, networks of width $d_y - 1$ are not dense in both $L^p(\mathcal{K}, \mathbb{R}^{d_y})$ and $C(\mathcal{K}, \mathbb{R}^{d_y})$.*

*Proof.* In this proof, we show that networks of width $d_y - 1$ are not dense in $L^p([0, 1]^{d_x}, \mathbb{R}^{d_y})$, which can be easily generalized to the cases of $L^p(\mathcal{K}, \mathbb{R}^{d_y})$ and $C(\mathcal{K}, \mathbb{R}^{d_y})$. In particular, we prove that there exist a continuous function $f^* \in L^p([0, 1]^{d_x}, \mathbb{R}^{d_y})$ and $\varepsilon > 0$ such that for any network $f$ of width $d_y - 1$, it holds that

$$\|f^* - f\|_p > \varepsilon.$$

Let $\Delta$ be a $d_y$-dimensional regular simplex with sidelength $\sqrt{2}$, that is isometrically embedded into $\mathbb{R}^{d_y}$. The volume of this simplex is given as $\mathrm{Vol}_{d_y}(\Delta) = \frac{\sqrt{d_y + 1}}{d_y!}$.[9] We denote the vertices of this simplex by $\{v_0, \ldots, v_{d_y}\}$. Then, we can construct the counterexample as follows.

$$f^*(x) = \begin{cases} v_i & \text{if } x_1 \in \left[\frac{2i}{2d_y + 1}, \frac{2i+1}{2d_y + 1}\right] \text{ for some } i \\ (2d_y + 1)(v_{i+1} - v_i)x_1 + (2i + 2)v_i - (2i + 1)v_{i+1} & \text{if } x_1 \in \left[\frac{2i+1}{2d_y + 1}, \frac{2i+2}{2d_y + 1}\right] \text{ for some } i \end{cases}.$$

In other words, $f^*(x)$ travels the vertices of $\Delta$ sequentially as we move $x_1$ from 0 to 1, staying at each vertex over an interval of length $\frac{1}{2d_y + 1}$ and traveling between vertices at a constant speed otherwise, i.e., $f^*$ is continuous and in $L^p([0, 1]^{d_x}, \mathbb{R}^{d_y})$.

Recalling (1), one can notice that any neural network $f$ of width less than $d_y$ and $L \geq 2$ layers can be decomposed as $t_L \circ g$, where $t_L : \mathbb{R}^k \to \mathbb{R}^{d_y}$ is the last affine transformation and $g$ denotes all the preceding layers, i.e., $g = \sigma_{L-1} \circ t_{L-1} \circ \cdots \circ \sigma_1 \circ t_1$. Here, we consider $k = d_y - 1$ as it suffices to cover cases $k < d_y - 1$. Now, we proceed as

$$\|f^* - f\|_p = \left(\int_{[0,1]^{d_x}} \|f^*(x) - f(x)\|_p^p \, dx\right)^{\frac{1}{p}}$$

$$\geq \left(\int_{[0,1]^{d_x}} \|f^*(x) - t_L \circ g(x)\|_p^p \, dx\right)^{\frac{1}{p}}$$

$$\geq \left(\int_{[0,1]^{d_x}} \inf_{u^*(x) \in \mathbb{R}^{d_y - 1}} \|f^*(x) - t_L(u^*(x))\|_p^p \, dx\right)^{\frac{1}{p}}$$

$$\geq \left(\frac{1}{2d_y + 1}\right)^{\frac{1}{p}} \inf_{t: \text{ affine map}} \max_{i \in [d_y + 1]} \inf_{u_i^* \in \mathbb{R}^{d_y - 1}} \|v_i - t(u_i^*)\|_p$$

$$\geq \left(\frac{1}{2d_y + 1}\right)^{\frac{1}{p}} \inf_{\mathcal{H} \in \mathfrak{H}} \max_{i \in [d_y + 1]} \inf_{a_i \in \mathcal{H}} \|v_i - a_i\|_p,$$

where $\mathfrak{H}$ denotes the set of all $(d_y - 1)$-dimensional hyperplanes in $\mathbb{R}^{d_y}$ and $[d_y + 1] := \{0, 1, \ldots, d_y\}$. As the vertices of $\Delta$ are $d_y + 1$ distinct points in a general position, $\inf_{\mathcal{H} \in \mathfrak{H}} \max_{i \in [d_y + 1]} \inf_{a_i \in \mathcal{H}} \|v_i - a_i\|_p > 0$. To make this argument more concrete, we take a volumetric approach; for any $k$-dimensional hyperplane $\mathcal{H} \in \mathbb{R}^{d_y}$, we have

$$\mathrm{Vol}_{d_y}(\Delta) \leq 2 \cdot \mathrm{Vol}_{d_y - 1}(\pi_{\mathcal{H}}(\Delta)) \cdot \max_{i \in [d_y + 1]} \inf_{a_i \in \mathcal{H}} \|v_i - a_i\|_2,$$

---

[9] $\mathrm{Vol}_d(\mathcal{S})$ denotes the volume of $\mathcal{S}$ in the $d$-dimensional Euclidean space.

where $\pi_{\mathcal{H}}$ denotes the projection onto $\mathcal{H}$. As projection is contraction and the distance between any two points are at most $\sqrt{2}$, it holds that for any $\mathcal{H}$,

$$\max_{i \in [d_y+1]} \inf_{a_i \in \mathcal{H}} \|v_i - a_i\|_2 \geq \frac{\text{Vol}_{d_y}(\Delta)}{2 \cdot \text{Vol}_{d_y-1}\left(\{x \in \mathbb{R}^{d_y-1} : \|x\|_2 \leq 1\}\right)}$$

$$= \frac{\sqrt{d_y+1}}{2 \cdot d_y!} \cdot \Gamma\left(\frac{d_y+1}{2}\right) \cdot \left(\frac{2}{\pi}\right)^{\frac{d_y-1}{2}}$$

where $\Gamma$ denotes the gamma function, and we use the fact that $\text{Vol}_{d_y-1}\left(\{x \in \mathbb{R}^{d_y-1} : \|x\|_2 \leq 1\}\right) \geq \text{Vol}_{d_y-1}(\pi_{\mathcal{H}}(\Delta))$ as $\Delta$ can be contained in a $d_y$-dimensional unit ball, and hence $\pi_{\mathcal{H}}(\Delta)$ can be contained in a $(d_y - 1)$-dimensional unit ball. Thus we have $\|f^* - f\|_p > \varepsilon$ with

$$\varepsilon = \frac{d_y^{\frac{1}{p}-\frac{1}{2}}\sqrt{d_y+1}}{2 \cdot (2d_y+1)^{\frac{1}{p}} \cdot d_y!} \cdot \Gamma\left(\frac{d_y+1}{2}\right) \cdot \left(\frac{2}{\pi}\right)^{\frac{d_y-1}{2}},$$

for $p \geq 2$ and

$$\varepsilon = \frac{\sqrt{d_y+1}}{2 \cdot (2d_y+1)^{\frac{1}{p}} \cdot d_y!} \cdot \Gamma\left(\frac{d_y+1}{2}\right) \cdot \left(\frac{2}{\pi}\right)^{\frac{d_y-1}{2}},$$

for $p < 2$. This completes the proof of Lemma 18. □

### B.2 PROOF OF TIGHT LOWER BOUND IN THEOREM 3

In this section, we prove the tight lower bound in Theorem 3. Since we already have the width-$d_y$ lower bound by Lemma 18 and it is already proven that RELU networks of width $d_x$ is not dense in $C(\mathcal{K}, \mathbb{R}^{d_y})$ (Hanin and Sellke, 2017), we prove the tight lower bound in Theorem 3 by showing the following statement: There exist $f^* \in C([0,1]^{d_x}, \mathbb{R})$ and $\varepsilon > 0$ such that for any RELU+STEP network $f$ of width $d_x$ containing at least one STEP, it holds that

$$\|f^* - f\|_\infty > \varepsilon.$$

Without loss of generality, we assume that $f$ has $d_x$ hidden neurons at each layer except for the output layer and all affine transformations in $f$ are invertible (see Section 5.1).

Our main idea is to utilize properties of *level sets* of width-$d_x$ RELU+STEP networks (Hanin and Sellke, 2017) defined as follows: Given a network $f$ of width $d_x$, we call a connected component of $f^{-1}(y)$ for some $y$ as a level set. Level sets of RELU+STEP networks have a property described by the following lemma. We note that the statement and the proof of Lemma 19 is motivated by Lemma 6 of (Hanin and Sellke, 2017).

**Lemma 19.** *Let $f$ be a RELU+STEP network of width $d_x$ containing at least one STEP. Then, for any level set $\mathcal{S}$ of $f$, $\mathcal{S}$ is unbounded unless it is empty.*

*Proof of Lemma 19.* Let $\ell^*$ be the smallest number such that STEP appears at the $\ell^*$-th layer. In this proof, we show that all level sets of the first $\ell^*$ layers of $f$ are either unbounded or empty. Then the claim of Lemma 19 directly follows. We prove this using the mathematical induction on $\ell^*$. Recalling (1), we denote by $f_\ell$ the mapping of the first $\ell$ layers of $f$:

$$f_\ell := \sigma_\ell \circ t_\ell \circ \cdots \circ \sigma_1 \circ t_1.$$

First, consider the base case: $\ell^* = 1$. Assume without loss of generality that the activation function of the first hidden node in $\sigma_1$ is STEP. Then for any $x$, the STEP activation maps the first component of $t_1(x)$ to 1 if $(t_1(x))_1 \geq 0$, and to 0 otherwise. This means that there exists a ray $\mathcal{R}$ starting from $x$ such that $f_1(\mathcal{R}) = \{f_1(x)\}$. Hence, any level set of $f_1$ is either unbounded or empty.

Now, consider the case that $\ell^* > 1$. Then, until the $(\ell^* - 1)$-th layer, the network only utilizes RELU. Here, the level sets of $f_{\ell^*-1}$ can be characterized using the following lemma.

**Lemma 20** [Hanin and Sellke (2017, Lemma 6)]. *Given a RELU network $g$ of width $d_x$, let $\mathcal{S} \subset \mathbb{R}^{d_x}$ be a set such that $x \in \mathcal{S}$ if and only if inputs to all RELU in $g$ are strictly positive, when computing $g(x)$. Then, $\mathcal{S}$ is open and convex, $g$ is affine on $\mathcal{S}$, and any bounded level set of $g$ is contained in $\mathcal{S}$.*

Consider $\mathcal{S}$ of $f_{\ell^*-1}$ as in Lemma 20 and consider a level set $\mathcal{T}$ of $f_{\ell^*}$ containing some $x$, i.e., $\mathcal{T} \neq \emptyset$. If $x \notin \mathcal{S}$, then $\mathcal{T}$ is unbounded by Lemma 20. If $x \in \mathcal{S}$, we argue as the base case. The preimage of $f_{\ell^*}(x)$ of the $\ell^*$-th layer (i.e., $\sigma \circ t_{\ell^*}$) contains a ray. If this ray is contained in $f_{\ell^*-1}(\mathcal{S})$, then $\mathcal{T}$ is unbounded as $f_{\ell^*-1}$ is invertible and affine on $\mathcal{S}$. Otherwise, $\mathcal{T} \setminus \mathcal{S} \neq \emptyset$ and it must be unbounded as any level set of $f_{\ell^*-1}$ not contained in $\mathcal{S}$ is unbounded by Lemma 20. This completes the proof of Lemma 19. □

Now, we continue the proof of the tight lower bound in Theorem 3 based on Lemma 19. We note that our argument is also from the proof of the lower bound in Theorem 1 of (Hanin and Sellke, 2017).

Consider $f^* : [0, 1]^{d_x} \to \mathbb{R}$ defined as

$$f^*(x_1, \ldots, x_{d_x}) := \sum_{i=1}^{d_x} \left( x_i - \frac{1}{2} \right)^2.$$

Then, for $a = \frac{1}{4}$ and $b = 0$, one can observe that $(f^*)^{-1}(a)$ is a sphere of radius $\frac{1}{2}$ centered at $(f^*)^{-1}(b) = \{(\frac{1}{2}, \ldots, \frac{1}{2})\}$. Namely, any path from $(f^*)^{-1}(b)$ to infinity must intersect with $(f^*)^{-1}(a)$. Now, suppose that a RELU+STEP network $f$ of width $d_x$ satisfies that $\|f^* - f\|_\infty \leq \frac{1}{16}$. Then, the level set of $f$ containing $(\frac{1}{2}, \ldots, \frac{1}{2})$ must be unbounded by Lemma 19, and hence, must intersect with $(f^*)^{-1}(a)$. However, as $f^* \circ (f^*)^{-1}(a) = \frac{1}{4}$ and $f^* \circ (f^*)^{-1}(b) = 0$, this contradicts with $\|f^* - f\|_\infty \leq \frac{1}{16}$. This completes the proof of the tight lower bound $\max\{d_x + 1, d_y\}$ in Theorem 3.

### B.3   PROOF OF TIGHT LOWER BOUND IN THEOREM 1

In this section, we prove the tight lower bound in Theorem 1. Since we already have the $d_y$ lower bound by Lemma 18, we prove the tight lower bound in Theorem 1 by showing the following statement: There exist $f^* \in L^p(\mathbb{R}^{d_x}, \mathbb{R})$ and $\varepsilon > 0$ such that for any continuous function $f$ represented by a RELU network of width $d_x$, it holds that

$$\|f^* - f\|_\infty > \varepsilon.$$

Note that this statement can be easily generalized to an arbitrary codomain. To derive the statement, we prove a stronger statement: For any RELU network $f$ of width $d_x$, either

$$f \notin L^p(\mathbb{R}^{d_x}, \mathbb{R}) \quad \text{or} \quad f = 0 \tag{10}$$

where $f = 0$ denotes that $f$ is a constant function mapping any input to zero. Then it leads us to the desired result directly. Without loss of generality, we assume that $f$ has $d_x$ hidden neurons at each layer except for the output layer and all affine transformations in $f$ are invertible (see Section 5.1).

As in the proof of the tight upper bound in Theorem 3, we utilize properties of level sets of $f$ given by Lemma 20. Let $\mathcal{S}$ be a set defined in Lemma 20 of $f$. By the definition of $\mathcal{S}$, one can observe that $\mathbb{R}^{d_x} \setminus \mathcal{S} \neq \emptyset$. Then, a level set $\mathcal{T}$ containing some $x \in \mathbb{R}^{d_x} \setminus \mathcal{S}$ must be unbounded by Lemma 20. Here, if $y := f(x) > 0$, then for $\delta := \omega_f^{-1}(\frac{y}{2})$, we have $f(x') \geq \frac{y}{2}$ for all

$$x' \in \mathcal{T}' := \{x' \in \mathbb{R}^{d_x} : \exists x \in \mathcal{T} \text{ such that } \|x' - x\|_\infty \leq \delta\}.$$

Since $\mathcal{T}'$ contains $\mathcal{T}$ which is an unbounded set, one can easily observe that $\mu(\mathcal{T}') = \infty$ and hence, $\int_{\mathcal{T}'} |f(x)|^p d(x) = \infty$, i.e., $f \notin L^p(\mathbb{R}^{d_x}, \mathbb{R})$.[10] One can derive the same result for $f(x) < 0$.

Suppose that $f(x) = 0$ for all $x \in \mathbb{R}^{d_x} \setminus \mathcal{S}$. Then, $f(x) = 0$ for all $x \in \partial \mathcal{S}$ as $\mathcal{S}$ is open (see Lemma 20). Furthermore, we claim that $f(\mathcal{S}) = \{0\}$ or $f \notin L^p(\mathbb{R}^{d_x}, \mathbb{R})$. For any $s \in \mathcal{S}$, consider any two rays of opposite directions starting from $s$. If one ray is contained in $\mathcal{S}$ and $f \in L^p(\mathbb{R}^{d_x}, \mathbb{R})$, then its image for $f$ must be $\{0\}$. If the image of $f$ is not $\{0\}$, using the similar argument for the case $f(x) > 0$ leads us to $f \notin L^p(\mathbb{R}^{d_x}, \mathbb{R})$. Then, one can conclude that $f(s) = 0$. If both rays are not contained in $\mathcal{S}$, they must both intersect with $\partial \mathcal{S}$. Then, since $f$ is affine on $\mathcal{S}$, $f(s)$ must be zero as $f(\partial \mathcal{S}) = \{0\}$. Hence, we prove the claim.

This completes the proof of the tight upper bound in Theorem 1.

---

[10] $\mu$ denotes the Lebesgue measure.

### B.4 PROOF OF LEMMA 5

Let $\phi(x) = Ax + b$ for some invertible matrix $A = [a_1, a_2]^\top \in \mathbb{R}^{2\times 2}$ and for some vectors $a_1, a_2, b \in \mathbb{R}^2$. Then, it is easy to see that if

$$\langle a_1, x \rangle + b_1 \geq 0 \quad \text{and} \quad \langle a_2, x \rangle + b_2 \geq 0,$$

i.e., $x \in \mathcal{S}$, then $\phi^{-1} \circ \sigma \circ \phi(x) = x$. Hence, the first statement of Lemma 5 holds.

Now, consider the second statement of Lemma 5. Suppose that $\langle a_1, x \rangle + b_1 \geq 0$ but $\langle a_2, x \rangle + b_2 < 0$. Then, one can easily observe that $\phi^{-1} \circ \sigma \circ \phi$ maps a ray

$$\{x' \in \mathbb{R}^2 : \langle a_1, x' \rangle = \langle a_1, x \rangle, \langle a_2, x' \rangle + b_2 < 0\}$$

containing $x$ to a single point $\phi^{-1}\big((\langle a_1, x \rangle + b_1, 0)\big)$, which is on $\partial \mathcal{S}$. In addition, similar arguments hold for cases that $\langle a_1, x \rangle + b_1 < 0, \langle a_2, x \rangle + b_2 \geq 0$ and $\langle a_1, x \rangle + b_1 < 0, \langle a_2, x \rangle + b_2 < 0$. This completes the proof of Lemma 5.

### B.5 PROOF OF LEMMA 6

We first prove the first statement of Lemma 6 using the proof by contradiction. Suppose that $x' = x$ and $x' \notin \mathcal{T}'$ but $x'$ is not in a bounded path-component of $\mathbb{R}^2 \setminus \mathcal{T}'$. Here, note that $x = x' \in \mathcal{S}$ for $\mathcal{S}$ defined in Lemma 5. Then, there exists a path $\mathcal{P}$ from $x'$ to infinity such that $\mathcal{P} \cap \mathcal{T}' = \emptyset$. If $\mathcal{P} \subset \text{int}(\mathcal{S})$[11], then the preimages of $\mathcal{P}$ and $\mathcal{T}' \cap \text{int}(\mathcal{S})$ under $\phi^{-1} \circ \sigma \circ \phi$ stay identical to their corresponding images, i.e., $\mathcal{P}$ and $\mathcal{T}' \cap \text{int}(\mathcal{S})$ (by Lemma 5). This contradicts the assumption that $x$ is in a bounded path-component of $\mathbb{R}^2 \setminus \mathcal{T}$. Hence, it must hold that $\mathcal{P} \not\subset \text{int}(\mathcal{S})$.

Let $x^* \notin \mathcal{T}'$ be the first point in $\mathcal{P} \cap \partial \mathcal{S}$ in the trajectory of $\mathcal{P}$ starting from $x'$. Then, the preimage of $x^*$ contains a ray $\mathcal{R}$ starting from $x^*$ (see the proof of Lemma 5 for the details) which must not intersect with $\mathcal{T}$; had the ray $\mathcal{R}$ intersected with $\mathcal{T}$, then $\mathcal{R} \cap \mathcal{T}$ must have mapped to $x^*$, which contradicts $x^* \notin \mathcal{T}'$ and the definition of $\mathcal{P}$. Furthermore, from the definition of $x^*$, the subpath $\mathcal{P}^\dagger$ of $\mathcal{P}$ from $x'$ to $x^*$ excluding $x^*$ satisfies $\mathcal{P}^\dagger \subset \text{int}(\mathcal{S})$. Hence, the preimages of $\mathcal{P}^\dagger$ and $\mathcal{T}' \cap \text{int}(\mathcal{S})$ under $\phi^{-1} \circ \sigma \circ \phi$ stay identical by Lemma 5. This implies that there exist a path $\mathcal{P}^\dagger$ from $x$ to $x^*$, and then a path $\mathcal{R}$ from $x^*$ to infinity, not intersecting with $\mathcal{T}$. This contradicts the assumption of Lemma 6. This completes the proof of the first statement of Lemma 6.

Now, consider the second statement of Lemma 6. By Lemma 5, $x \neq x'$ implies that $x \notin \mathcal{S}$ and $x' \in \partial \mathcal{S}$. Here, as the preimage of $x'$ contains a ray from $x'$ containing $x$, this ray must intersect with $\mathcal{T}$ from the assumption of Lemma 6. Hence, $x' \in \mathcal{T}'$ and this completes the proof of the second statement of Lemma 6.

By combining the proofs of the first and the second statements of Lemma 6, we complete the proof of Lemma 6.

### B.6 PROOF OF LEMMA 7

Before starting our proof, we first introduce the following definitions and lemma. The proof of Lemma 21 is presented in Appendix B.7.

**Definition 2.** *Definitions related to curves, loops, and polygons are listed as follows: For $\mathcal{U} \subset \mathbb{R}^2$ and $\mathcal{F}(\mathcal{U}) := \{f \in C([0,1], \mathbb{R}^2) : f([0,1]) = \mathcal{U}\}$,*

- *$\mathcal{U}$ is a "curve" if there exists $f \in \mathcal{F}(\mathcal{U})$.*

- *$\mathcal{U}$ is a "simple curve" if there exists an injective $f \in \mathcal{F}(\mathcal{U})$.*

- *$\mathcal{U}$ is a "loop" if there exists $f \in \mathcal{F}(\mathcal{U})$ such that $f(1) = f(0)$.*

- *$\mathcal{U}$ is a "simple loop" if there exists $f \in \mathcal{F}(\mathcal{U})$ such that $f(1) = f(0)$ and $f$ is injective on $[0,1)$.*

- *$\mathcal{U}$ is a "polygon" if there exists a piecewise linear $f \in \mathcal{F}(\mathcal{U})$ such that $f(1) = f(0)$.*

---

[11]$\text{int}(\mathcal{S})$ denotes the interior of $\mathcal{S}$.

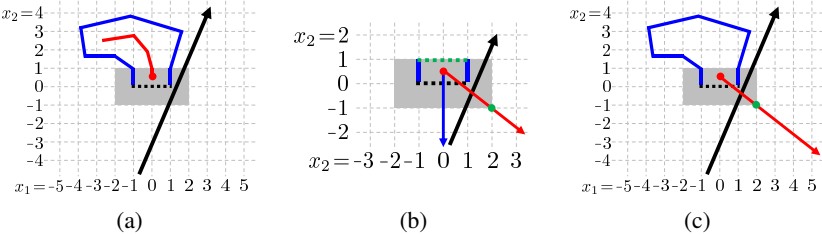

Figure 3: (a) Illustration of $\mathcal{U}$, $g_{\ell^*}(q)$. (b) Illustration of $\mathcal{V}$, $g_{\ell^*}(q)$, $v$. (c) Illustration of $\mathcal{U}$, $g_{\ell^*}(q)$, $v$.

- $\mathcal{U}$ is a "simple polygon" if there exists a piecewise linear $f \in \mathcal{F}(\mathcal{U})$ such that $f(1) = f(0)$ and $f$ is injective on $[0, 1)$.

**Lemma 21.** *Suppose that $g_{\ell^*}([0, p_1]) \cap g_{\ell^*}([p_2, 1]) = \emptyset$ and $g_{\ell^*}([0, p_1]) \setminus \mathcal{B}$ is contained in a bounded path-component of $\mathbb{R}^2 \setminus \mathcal{U}$ for some $\mathcal{U} \subset g_{\ell^*}([p_2, 1]) \cup \mathcal{B}$. Then, $g_{\ell^*}([0, p_1]) \setminus \mathcal{B}$ is contained in a bounded path-component of $\mathbb{R}^2 \setminus (g_{\ell^*}([p_2, 1]) \cup \mathcal{B})$.*

In this proof, we prove that if $g_{\ell^*}([0, p_1]) \cap g_{\ell^*}([p_2, 1]) = \emptyset$, then $g_{\ell^*}([0, p_1]) \setminus \mathcal{B}$ is contained in a bounded path-component of $\mathbb{R}^2 \setminus \mathcal{U}$ for some simple polygon $\mathcal{U} \subset g_{\ell^*}([p_2, 1]) \cup \mathcal{B}$. Then, the statement of Lemma 7 directly follows by Lemma 21.

To begin with, consider a loop $g_{\ell^*}([p_2, 1]) \cup \mathcal{L}(g_{\ell^*}(p_2), g_{\ell^*}(1))$ where $\mathcal{L}(x, x')$ denotes the line segment from $x$ to $x'$, i.e.,

$$\mathcal{L}(x, x') := \{\lambda \cdot x + (1 - \lambda) \cdot x' : \lambda \in [0, 1]\}.$$

Then, the loop consists of a finite number of line segments, as an image of an interval of a RELU network is piecewise linear as well as $\mathcal{L}(g_{\ell^*}(p_2), g_{\ell^*}(1))$, i.e., the loop is a polygon.

Since $g_{\ell^*}([p_2, 1]) \cup \mathcal{L}(g_{\ell^*}(p_2), g_{\ell^*}(1))$ consists of line segments, under the assumption $\|f^* - f\|_\infty \leq \frac{1}{100}$, one can easily construct a simple loop $\mathcal{U}$ in $g_{\ell^*}([p_2, 1]) \cup \mathcal{L}(g_{\ell^*}(p_2), g_{\ell^*}(1))$ so that $\mathcal{U}$ contains simple curves from the midpoint of $\mathcal{L}(g_{\ell^*}(p_2), g_{\ell^*}(1))$ to a point near the point $(-1, 1)$, then to a point near the point $(1, 1)$, and finally to the midpoint of $\mathcal{L}(g_{\ell^*}(p_2), g_{\ell^*}(1))$. We note that $\mathcal{U}$ also consists of line segments, i.e., $\mathcal{U}$ is a simple polygon. Figure 3(a) illustrates $\mathcal{U}$ where line segments from $g_{\ell^*}([p_2, 1])$ is drawn in blue and line segments from $L(g_{\ell^*}(p_2), g_{\ell^*}(1))$ indicated by dotted black line.

Now, choose $q \in (0, p_1)$ such that $f^*(q) = (0, \frac{1}{2})$. Since $\|f^* - f\|_\infty \leq \frac{1}{100}$ and by the definition of $\ell^*$,

$$f(q) = g_{\ell^*}(q) \in \{x \in \mathbb{R}^2 : \|x - (0, \tfrac{1}{2})\|_\infty \leq \tfrac{1}{100}\}$$

which is illustrated by the red dot in Figure 3. Then, we claim the following statement:

$$g_{\ell^*}(q) \text{ is contained in a bounded path-component of } \mathbb{R}^2 \setminus \mathcal{U}. \tag{11}$$

From the definition of $q$ and the path-connectedness of $g_{\ell^*}([0, q])$, one can observe that proving the claim (11) leads us to that $g_{\ell^*}([0, q])$ is contained in a bounded path-component of $\mathbb{R}^2 \setminus \mathcal{U}$ unless $\mathcal{U} \cap g_{\ell^*}([0, q]) \neq \emptyset$. Since $g_{\ell^*}([0, p_1]) \setminus \mathcal{B} \subset g_{\ell^*}([0, q])$ by the definitions of $q, \ell^*$ and the assumption that $\|f^* - f\|_\infty \leq \frac{1}{100}$, this implies that if $g_{\ell^*}([0, p_1]) \cap g_{\ell^*}([p_2, 1]) = \emptyset$, then $g_{\ell^*}([0, p_1]) \setminus \mathcal{B}$ is contained in a bounded path-component of $\mathbb{R}^2 \setminus \mathcal{U}$. Hence, (11) implies the statement of Lemma 7.

**Proof of claim (11).** To prove the claim (11), we first introduce the following lemma.

**Lemma 22** [Jordan curve theorem (Tverberg, 1980)]. *For any simple loop $\mathcal{O} \subset \mathbb{R}^2$, $\mathbb{R}^2 \setminus \mathcal{O}$ consists of exactly two path-components where one is bounded and another is unbounded.*

Lemma 22 ensures the existence of a bounded path-component of $\mathbb{R}^2 \setminus \mathcal{U}$.

Furthermore, to prove the claim (11), we introduce the parity function $\pi_\mathcal{U} : \mathbb{R}^2 \setminus \mathcal{U} \to \{0, 1\}$: For $x \in \mathbb{R}^2 \setminus \mathcal{U}$ and a ray starting from $x$, $\pi_\mathcal{U}(x)$ counts the number of times that the ray "properly"

intersects with $\mathcal{U}$ (reduced modulo 2) where the proper intersection is an intersection where $\mathcal{U}$ enters and leaves on different sides of the ray. Here, it is well-known that $\pi_{\mathcal{U}}(x)$ does not depend on the choice of the ray, i.e., $\pi_{\mathcal{U}}$ is well-defined. We refer the proof of Lemma 2.3 by Thomassen (1992) and the proof of Lemma 1 by Tverberg (1980) for more details. Here, $\pi_{\mathcal{U}}$ characterizes the "position" of $x$ as $\pi_{\mathcal{U}}(x) = 0$ if and only if $x$ is in the unbounded path-component of $\mathbb{R}^2 \setminus \mathcal{U}$, which is known as the even-odd rule (Shimrat, 1962; Hacker, 1962). Hence, proving that $\pi_{\mathcal{U}}(g_{\ell^*}(q)) = 1$ would complete the proof of the claim (11).

Recall that there exists the line (e.g., the black arrow in Figure 3) that intersects with $\mathcal{B}$ and the image of $g_{\ell^*}$ can be at only "one side" of the line (see Section 5.2 for details). Since $\mathcal{B}$ is open, there exists a "vertex" $v \in \partial\mathcal{B}$ (e.g., the green dot in Figures 3(b) and 3(c)) such that $v$ is in the "other side" of the line.[12] We prove $\pi_{\mathcal{U}}(g_{\ell^*}(q)) = 1$ by counting the number of proper intersections between the ray $\mathcal{R}$ from $g_{\ell^*}(q)$ passing through $v$ (the red arrow in Figures 3(b) and 3(c) illustrates $\mathcal{R}$).

To simplify showing $\pi_{\mathcal{U}}(g_{\ell^*}(q)) = 1$, we consider two points $z_1, z_2 \in \mathcal{U} \cap \partial\mathcal{B}$ near the points $(-1, 1), (1, 1)$, respectively, such that the simple curve $\mathcal{P}$ in $\mathcal{U}$ from $z_1$ to $z_2$ is contained in $\mathcal{B}$ except for $z_1, z_2$. Then, one can observe that $\mathcal{P}$ and $\mathcal{L}(z_1, z_2)$ forms a simple loop which we call $\mathcal{V}$. Figure 3(b) illustrates $\mathcal{V}$ where the black dotted line indicates the line segment from $\mathcal{L}\big(g_{\ell^*}(p_2), g_{\ell^*}(1)\big)$, the blue line indicates the line segments from $g_{\ell^*}([p_2, 1])$, and the green dotted line indicates $\mathcal{L}(z_1, z_2)$; from the definition of $\mathcal{P}$, the blue and green lines together correspond to $\mathcal{P}$.

Then, $\pi_{\mathcal{V}}(g_{\ell^*}(q)) = 1$ as a ray from $g_{\ell^*}(q)$ of the downward direction (the blue arrow in Figure 3(b)) only properly intersects once with $\mathcal{V}$ at some point in $\mathcal{L}\big(g_{\ell^*}(p_2), g_{\ell^*}(1)\big)$, under the assumption that $\|f^* - f\|_\infty \le \frac{1}{100}$. From the property of $\pi_{\mathcal{V}}$, this implies that the ray $\mathcal{R}$ starting from $g_{\ell^*}(q)$ and passing through $v$ (e.g., the red arrow in Figures 3(b) and 3(c)) must properly intersect with $\mathcal{V}$ odd times. Furthermore, from the construction of $\mathcal{U}$ and $\mathcal{V}$, definition of $\ell^*$, and under the assumption that $\|f^* - f\|_\infty \le \frac{1}{100}$, one can observe that the simple curve in $\mathcal{U}$ from $z_1$ to $z_2$ *not* contained in $\mathcal{B}$ (i.e., $\mathcal{U} \setminus \mathcal{P}$) can only intersect with $\mathcal{B}$ within the $\ell_\infty$ balls of radius $\frac{2}{100}$ centered at the points $(-1, 1)$ and $(1, 1)$. This is because if $\mathcal{U} \setminus \mathcal{P}$ intersects with $\mathcal{B}$ outside these $\ell_\infty$ balls, then by definition of $\ell^*$, the network cannot make further modifications in $\mathcal{B}$, hence contradicting the approximation assumption $\|f^* - f\|_\infty \le \frac{1}{100}$. In other words, all proper intersections between $\mathcal{U}$ and $\mathcal{R}$ are *identical* to those between $\mathcal{V}$ and $\mathcal{R}$. This implies that $\pi_{\mathcal{U}}(g_{\ell^*}(q)) = 1$ and hence, $g_{\ell^*}(q)$ is in the bounded path-component of $\mathbb{R}^2 \setminus \mathcal{U}$. This completes the proof of the claim (11) and therefore, completes the proof of Lemma 7.

### B.7 PROOF OF LEMMA 21

Suppose that $g_{\ell^*}([0, p_1]) \cap g_{\ell^*}([p_2, 1]) = \emptyset$ and $g_{\ell^*}([0, p_1]) \setminus \mathcal{B}$ is contained in a bounded path-component of $\mathbb{R}^2 \setminus \mathcal{U}$ for some $\mathcal{U} \subset g_{\ell^*}([p_2, 1]) \cup \mathcal{B}$. If $g_{\ell^*}([0, p_1]) \setminus \mathcal{B}$ is path-connected, then the statement of Lemma 21 directly follows. Hence, suppose that $g_{\ell^*}([0, p_1]) \setminus \mathcal{B}$ has more than one path-components. To help the proof, we introduce the following Lemma.

**Lemma 23.** *If $g_{\ell^*}(p) \in \partial\mathcal{B}$ for some $p \in [0, 1]$, then $f(p) = g_{\ell^*}(p)$.*

*Proof of Lemma 23.* Suppose that $f(p) \ne g_{\ell^*}(p)$. Then, $\phi_\ell^{-1} \circ \sigma \circ \phi_\ell(g_{\ell^*}(p)) \ne g_{\ell^*}(p)$ for some $\ell > \ell^*$. By Lemma 5, there exist $a_1, a_2 \in \mathbb{R}^2$ and $b_1, b_2 \in \mathbb{R}$ such that $\phi_\ell^{-1} \circ \sigma \circ \phi_\ell(x) = x$ if and only if $\langle a_1, x \rangle + b_1 \ge 0, \langle a_2, x \rangle + b_2 \ge 0$. Without loss of generality, we assume that $\langle a_1, g_{\ell^*}(p) \rangle + b_1 < 0$. Since $g_{\ell^*}(p) \in \partial\mathcal{B}$, there exists $z \in \mathcal{B}$ such that $\langle a_1, z \rangle + b_1 < 0$, i.e., $\phi_\ell^{-1} \circ \sigma \circ \phi_\ell(z) \ne z$, which contradicts to the definition of $\ell^*$ by Lemma 5. This completes the proof of Lemma 23 □

By Lemma 23 and the assumption that $\|f^* - f\| \le \frac{1}{100}$, $g_{\ell^*}([0, p_1]) \setminus \mathcal{B}$ can only intersect with $\partial\mathcal{B}$ within the $\ell_\infty$ ball $\mathcal{O}$ of radius $\frac{2}{100}$ centered at the point $(0, 1)$. Hence, all path-components of $g_{\ell^*}([0, p_1]) \setminus \mathcal{B}$ intersect with the line segment $\partial\mathcal{B} \cap \mathcal{O}$. In other words, $g_{\ell^*}([0, p_1]) \setminus \mathcal{B}$ is in a path-component of $\mathbb{R}^2 \setminus (g_{\ell^*}([p_2, 1]) \cup \mathcal{B})$ unless $g_{\ell^*}([p_2, 1])$ intersects with $\partial\mathcal{B} \cap \mathcal{O}$. However, by Lemma 23 and the assumption that $\|f^* - f\| \le \frac{1}{100}$, $g_{\ell^*}([p_2, 1])$ must not intersect with $\partial\mathcal{B} \cap \mathcal{O}$. This completes the proof of Lemma 21.

---

[12]A vertex denotes one of the points $(2, -1), (2, 1), (-2, -1), (-2, 1)$.

