# OpenReview forum: "Minimum Width for Universal Approximation"
_ICLR.cc/2021/Conference — ICLR 2021 Spotlight_

### Official Review · AnonReviewer1 · 2020-10-23
**Tight bounds on minimum width needed by ReLU networks for approximation**

**Rating:** 8
**Confidence:** 4

**Review:**

This paper studies the problem of universal approximation with networks of bounded width and arbitrary depth. The objective is to understand what's the minimum width necessary to approximate any function in a suitable space. The main results of the paper can be summarized as follows:

(1) ReLU networks approximate functions in $L^p(\mathbb R^{d_x}, \mathbb R^{d_y})$ if and only if the width is at least $\max(d_x+1, d_y)$. This result is tight and improves upon (Kidger and Lyons, 2020), which give an upper bound of $d_x+d_y+1$.

(2) The same result does *not* hold if we look at the approximation in $C(K, \mathbb R^{d_y})$, $K$ being a compact. The author(s) prove that the minimum width is 3 by giving a counterexample.

(3) In order to maintain the width of $\max(d_x+1, d_y)$ in $C(K, \mathbb R^{d_y})$, it suffices to use ReLU and threshold activations.

(4) An upper bound on the width of $\max(d_x+2, d_y+1)$ for approximation in $L^p(K, \mathbb R^{d_y})$ is given for a wide class of activation functions.


The proofs contain two main elements of novelty:

(a) The upper bounds (results (1)-(3)-(4) above) rely on what the author(s) call a 'coding scheme'. The input is mapped to a one-dimensional codeword by an encoder; the codeword is mapped to a one-dimensional target by a memorizer; and the target is mapped to vector close to the output by the decoder. The idea is that all these three maps (encoder, memorizer and decoder) can be constructed with neural networks of width $\max(d_x+1, d_y)$ by using ideas of prior results, e.g., (Hanin and Sellke, 2017). The point of the coding scheme is to decouple the input and the output dimension: the original mapping from a space of dimension $d_x$ to a space of dimension $d_y$ is broken into (i) a mapping from dimension $d_x$ to dimension 1, (ii) a mapping from dimension 1 to dimension 1, and (iii) a mapping from dimension 1 to dimension $d_y$. This is what allows to improve the bound on the width from  $d_x+d_y+1$ to $\max(d_x+1, d_y)$.

(b) The counterexample on which the lower bound is based (result (2) above) comes from a topological argument.

The paper is well written, the results are interesting and strong, the proof techniques are novel. Thus, I am generally positive about the submission.

I have a few questions/remarks:

(Q1) This is a general question. The author(s) provide a general upper bound on the width of $\max(d_x+2, d_y+1)$. Is that tight for a sub-class of activations? Any comment on how to improve it beyond ReLU?

(Q2) Lemma 5. Does the set $\mathcal S$ (or, equivalently, the choice of $a_1$, $a_2$, $b_1$, $b_2$) depend on $\phi$? It looks like this is the case, and it would be better to clarify this point.

(Q3) Section 5.2. It is mentioned that "by the definition of $\ell^*$ and Lemma 5, there exists a line intersecting with $\mathcal B$". How do you guarantee the intersection with $\mathcal B$? The set $\mathcal S$ in Lemma 5 is generic (and, in principle, may not give rise to an intersection).

(Q4) In Lemma 6, the author(s) talk about a bounded path-connected component without defining what path-connected means.

I spotted a couple of typos:

(T1) Page 6. "memorizer_{K, M}" should be "memorize_{K, M}".

(T2) "differentiable at at least". This occurs several times in the main text and the appendix as well.

---

> ### Author Response · Authors · 2020-11-22
> **Response to Reviewer 1**
>
> We sincerely appreciate your valuable comments, efforts, and time. We updated the draft according to the reviewers' comments and colored updated parts blue. We address each of your comments in detail as follows.
>
> **Tightness of width $\max\{d_x+2,d_y+1\}$ for general activation functions.**
> As the reviewer points out, our paper does not prove the tightness of $w_{\min}\le\max\{d_x+2,d_y+1\}$ for approximating $f\in L^p([0,1]^{d_x},\mathbb R^{d_y})$ using general activations. We first note that (Hanin and Sellke, 2017) shows $w_{\min}\le d_x+d_y$ for approximating $C(\mathcal K,\mathbb R^{d_y})$ with ReLU networks, which is smaller than $w_{\min}\le d_x+d_y+1$ in (Kidger and Lyons, 2020); so there is a difference by one neuron in the two results. In light of this difference, our upper bound $w_{\min}\le\max\{d_x+2,d_y+1\}$ can be improved to $w_{\min}\le\max\{d_x+1,d_y\}$ utilizing our encoding-decoding framework, if one can prove $w_{\min}\le d_x+d_y$ for approximating $f \in C(\mathcal K,\mathbb R^{d_y})$ with some activation functions. In this case, the upper bound can be shown to be tight because the lower bound $w_{\min}\ge\max\{d_x+1,d_y\}$ holds for general activation functions. We believe that proving the tightness of $w_{\min}\le\max\{d_x+2,d_y+1\}$ or improving $w_{\min}\le\max\{d_x+2,d_y+1\}$ is interesting future work.
>
>
> **Dependency of $\mathcal S$ on $\phi$ in Lemma 5.** We thank the reviewer for pointing this out. We explicitly mentioned the dependency of $\mathcal S$ on $\phi$ in Lemma 5 in the revised draft.
>
> **About the sentence "by the definition of $\ell^\*$ and Lemma 5..."** From the definition of $\ell^\*$, $f([0,1])\cap\mathcal B$ must have been constructed in the first $\ell^\*$ layers. Under this observation, the boundary of $\mathcal S$ in Lemma 5 determined by $\phi_{\ell^\*}$ must intersect with $\mathcal B$: If $\mathcal B\subset\mathcal S$, then $\phi_{\ell^\*}^{-1}\circ\sigma\circ\phi_{\ell^\*}(\mathcal B) = \mathcal B$ which contradicts the definition of $\ell^\*$ and if $\mathcal B\cap\mathcal S=\emptyset$, then $f([0,1])\cap\mathcal B$ cannot be constructed in the first $\ell^\*$ layers. In other words, there exists a line (e.g., a line containing one boundary of $\mathcal S$) intersecting with $\mathcal B$, such that the image $g_{\ell^\*}([0,1])$ lies in one side of the line. We revised the draft to make this point clearer.
>
> **Definition of path-component.** We thank the reviewer for pointing this out. We added the definition of the path-component (previously written as the 'path-connected component') in the revised draft as follows.
>
> "$\mathcal S\subset\mathcal T$ is a path-component of $\mathcal T$ if $\mathcal S$ is a maximal set satisfying the following condition: For any $x_1,x_2\in\mathcal S$, there exists a continuous function $f:[0,1]\rightarrow\mathcal S$ such that $f(0)=x_1$ and $f(1)=x_2$."
>
> **Typos.** We thank the reviewer for finding typos. We fixed them in the revised draft.

---

### Official Review · AnonReviewer3 · 2020-10-25
**Clean proofs, optimal bounds**

**Rating:** 7
**Confidence:** 3

**Review:**

This paper wants to establish tight theoretical lower-bounds on the minimum width required by a ReLU neural network to approximate "almost all" functions up to epsilon where the distance of approximation is defined using the Lp-norm. The paper improves the previously known bounds which lied in the range of d+1 and d+4 to exactly d+1. One of the results of this paper is to establish that this bound is exactly d+1 (I am summarizing the result while ignoring some precision/nuances). This result holds only when the distance is measured using the p-norm where p is finite. In the infinity norm setting, they show that this minimum width actually is indeed not a universal approximator. However, to ensure that the claimed minimum width also holds in the infinity-norm setting, they modify the activation functions to also allow for the threshold functions which is 1 if larger than 0 and 0 otherwise.


This is primarily a mathematical paper and the goal is to understand the minimum width required when arbitrary depth is allowed and thus, the results in this paper should be viewed as an interesting way to completely fill the full-understanding in the mathematics of deep neural nets. In general, the insights are unlikely to have any bearing on practice. With that disclaimer, here are my opinions of the paper. The paper provides a very interesting set of results. First, it fully solves the problem and gives exact bounds, where prior works have only given bounds that without getting the precise constants. Although one could argue that improving constants is not as significant, it still is important/interesting to learn that we can indeed find the precise constant (and arguably using a very nice proof/construction). Second, this paper also shows the dichotomy between approximation measured using Lp and L-infinity. In particular, there is no smooth transition between the minimum width required using only RELU activations between finite p and p-> infinity. However, it is also interesting that the fix to obtain the same minimum width is to allow for another additional activation function (and like RELU it is only non-differentiable at 0, Although, unlike RELU it is not continuous at 0 either). Overall the three set of results form an interesting and complete picture of the landscape for the required minimum width. The proof techniques involve viewing the mapping from input to output as a set of encoder, memorizer and decoder steps and implementing each of those using neural nets with RELU and optimal width. Combining the maximum of these widths gives the final required bound. I really like that the proof and construction is clean and simple to understand. Usually getting precise constants involves messy constructions; that is not the case in this paper and that is by far the biggest strength of this paper.

I had a few suggestions on this paper, that I feel could make the results in this paper even better.

- Even though the goal is to allow arbitrary depth, it is still instructive to add a discussion on the depth required in these bounds. In particular, what is the "cost" of having the width as small as the minimum required one, in terms of how deep the network should be, given a fixed error epsilon (or in terms of K, M if that is convenient)?


- In the result for p -> infinity, is the number of step activations required optimal? It is clear that you would need some step functions, but how many? Currently you seem to require one for every input coordinate. Is this optimal? A discussion of this and its optimality will make the theorem/result more stronger.

- It was not directly/quickly obvious to me how Lemma 8 + applying the q_K one for each input coordinate leads to a width of d+1 as opposed to 2d. The answer is that this is because you are mapping identity for all coordinates except one of them and repeating it d_x times. So stating this out, will make it quickly accessible to the reader. Even though this may seem "obvious", it improves the readability.

- There seems to be a caveat that the neural net uses the target function itself in the construction of its weights. Although this proof is about "information" needed, a discussion surrounding this and/or explaining why this is okay will again make the spirit of the results clear to the reader.

---

> ### Author Response · Authors · 2020-11-22
> **Response to Reviewer 3**
>
> We sincerely appreciate your valuable comments, efforts, and time. We updated the draft according to the reviewers' comments and colored updated parts blue. We address each of your comments in detail as follows.
>
> **Depth for achieving minimum width.** To address the reviewer's question, we added Section 4.5 in our revision with a discussion on the number of parameters in our constructions. Our ReLU+Step network construction achieving the tight minimum width for approximating $f\in C([0,1]^{d_x},\mathbb R^{d_y})$ in $\varepsilon$ error requires $O\big((\omega_{f}^{-1}(\Omega(\varepsilon)))^{-d_x}+1/\varepsilon\big)$ parameters, where $\omega_{f}$ denotes the modulus of continuity of $f$. Likewise, Our ReLU network construction achieving the tight minimum width for approximating $f\in L^p([0,1]^{d_x},\mathbb R^{d_y})$ in $\varepsilon$ error requires $O\big((\omega_{g}^{-1}(\Omega(\varepsilon)))^{-d_x}+1/\varepsilon\big)$ parameters where $g$ is some continuous function satisfying $\|f-g\|_p\le\varepsilon/2$. More details can be found in Section 4.5.
>
> **Optimal number of step functions.** As step activation functions are only used for constructing the encoder, our construction requires a number of step activation functions proportional to $d_x\cdot 2^K$, which will be, by our choice of $K$, equal to $O\big((\omega_{f^*}^{-1}(\Omega(\varepsilon))^{-1} \big)$ for approximating a target $f\in C([0,1]^{d_x},\mathbb R^{d_y})$ in $\varepsilon$ error (see Section 4.5 in the revised draft for details where we hide $d_x$ multiplicative factor in the big-O notation).
> However, unfortunately, we do not have proof that this is the minimum number of step activation functions for constructing universal approximators of the minimum width. We believe that identifying "how much discontinuity is necessary for universal approximators of the minimum width using ReLU activation functions" is an interesting future direction.
>
> **About applying $q_K$ to each input coordinate.** We thank the reviewer for pointing this out. We clarified this point in the revised draft as follows.
>
> "We sequentially apply $q_K$ to each input coordinate, by utilizing the extra width $1$ and Lemma 8. Here, when we apply $q_K$ to some coordinate (requiring width $2$), we preserve other coordinates by applying identity mapping (requiring width $d_x-1$) to maintain the width $d_x+1$."
>
> **Our neural network construction uses target function itself.**
> We are not completely sure if we understood the comment, but we will provide an answer to the following question: "Why does our network construction use the target function itself, while the target function is often unknown in practice?"
> If this question is not what the reviewer intended to ask, please do let us know.
>
> As the reviewer mentioned, our network construction utilizes the target function itself, which is not given in general. However, whenever a network tries to approximate a target function (whether the "full information" of the target function is available or not), it must have at least the "minimum size" (e.g., width, depth) sufficient for approximating the target function given the "full information." Namely, our results characterizing the exact minimum width for the universal approximation give us the fundamental lower bound on the width of networks for approximating target functions.
> We will add discussion on this in the revision if this is what the reviewer expected to discuss.

---

### Official Review · AnonReviewer2 · 2020-10-28

**Rating:** 7
**Confidence:** 4

**Review:**

==== Summary ====

The paper studies the minimal neural network width needed for universal approximation. While previous papers on the subject merely provided lower and upper bounds, this paper derives exact bounds on the minimal width. The paper considers both ReLU networks, as well as general activation functions, and examines approximation under both uniform norm and $L_p$ norm. In the case of functions with high-dimensional outputs, the derived bounds are better than previously thought ($d_x + d_y + 1$ to $\max(d_x + 1, d_y)$), with possible implications to practical  network design.

==== Detailed Review ====

Main strengths:
* The first exact bounds on the minimal width required for universal approximation, closing the gap between lower and upper bounds of previous results.
* Significant improvement for the case of $L_p$ functions with multi-dimensional output, with possible practical implications. For encoder-decoder style networks, which are common in many vision and language tasks, prior results had suggested a width double the input dimension, whereas here it has been demonstrated to be sufficient to use the same dimension as the input (+1).
* Show that uniform approximation with ReLU networks is slightly weaker than with networks with discontinuous activations (specifically, using both ReLU and step functions).
* The results are very clearly presented, as are their comparisons to previous bounds, including a very intuitive proof sketch.

Main weaknesses:
* For the case of scalar $L_p$ functions, the previous "gap" was merely off by one, i.e., $d_x + 1 \le w_{\min} \le d_x + 2$. In this particular situation, the improvement in absolute terms is quite minimal.
* Even in the general case, the prior lower and upper bounds were already asymptotically tight.

I recommend the paper be accepted, since the paper provides exact bounds that close the gap between lower and upper bounds, and that helps us understand these networks better. Although the gap has already been quite small in some situations, which might be regarded as incremental, in other cases, such as the very common encoder-decoder setting ($d_x = d_y$), the gap is large enough to affect practical considerations when designing compact networks. These improvements should be of interest to the general ML community.

---

> ### Author Response · Authors · 2020-11-22
> **Response to Reviewer 2**
>
> We sincerely appreciate your valuable comments, efforts, and time. We updated the draft according to the reviewers' comments and colored updated parts blue.
>
> We agree that the existing gap was small when $d_y$ is small and the previous upper bounds were already asymptotically tight. However, as the reviewer also mentioned, we completely close the gap while the existing gap ($d_x + 1$ vs. $d_x + d_y$) can be significant for networks having large output dimensions (e.g., encoder-decoder networks, super-resolution networks). We also believe that our exact characterization of the minimum width sheds light on our understanding of the expressive power of neural networks.

---

### Official Review · AnonReviewer4 · 2020-10-29
**This paper tightens the analysis of the minimum width for universal approximation with relu networks for Lp functions. The paper is well written and provides strong results.**

**Rating:** 7
**Confidence:** 4

**Review:**


Summary: The authors tighten the analysis of the minimum width for universal approximation with relu networks for Lp functions, where they have an exact characterization in terms of the input and output dimensions.

The paper is well written and easy to follow. The review of prior art is quite clear.

major comments/questions

1. It appears that the main result on Lp functions can be seen as a generalization of Hanin and Sellke (2017)'s result for d_y=1 to arbitrary d_y. Could you contrast the main differences in the proof technique?

2. It's interesting that uniform approximation is harder to obtain with relu networks, while relu+step works. However, step functions are not ideal from a practical perspective due to the a.e. zero gradient. Is there an optimization-friendly activation that satisfies uniform approximation with minimal width?

3. Does 'relu networks of width m' refer to networks where the number of layers is arbitrary? Could you please clarify?  What is the minimum number of layers required for the results to hold? Are there any implications for relu networks of fixed layers, e.g., two and three layers?

minor comments.
Table 1. Ours (Theorem 2) has max(d_x+1,d_y), but isn't d_y=2 in this case?

---

> ### Author Response · Authors · 2020-11-22
> **Response to Reviewer 4**
>
> We sincerely appreciate your valuable comments, efforts, and time. We updated the draft according to the reviewers' comments and colored updated parts blue. We address each of your comments in detail as follows.
>
> **Difference between result of Hanin and Sellke (2017).**
> Although our results are motivated by the result by Hanin and Sellke (2017), our results are not a generalization of the result by Hanin \& Sellke (2017).
>
> First, the target function class is different. While the result by Haninand Sellke (2017) aims to bound uniform error on a compact (i.e., bounded) domain denoted as $C(\mathcal K, \mathbb R)$ in our paper, our $L^p$ approximation result (Theorem 1) focuses on the entire $\mathbb R^{d_x}$, which requires to bound an integrated error in the unbounded domain ($L^p(\mathbb R^{d_x}, \mathbb R^{d_y})$ in our paper).
>
> Second, the proof strategy is different. Hanin and Sellke (2017) proved the sufficiency of width $d_x+d_y$ for approximating continuous functions in the uniform norm by (1) approximating a target continuous function by a max-min string and (2) implementing the max-min string by a ReLU network of width $d_x+d_y$.
> On the other hand, we propose an encoding-decoding framework and exactly construct the encoder, the memorizer, and the decoder (See Appendix A), and show the sufficiency of width $\max \{d_x + 1, d_y\}$. Note that the universal approximation result by Hanin and Sellke (2017) cannot exactly implement the encoder, the memorizer, and the decoder but can only approximate.
>
> We hope that our answer clarifies the main differences in the proof technique.
>
> **Optimization-friendly activation for uniform approximation with minimum width.** Step activation functions in our construction can be replaced by optimization-friendly activation functions. The key properties of the Step activation function in our construction are its discontinuity and flat regions around it. Hence, any activation function $\rho$ satisfying that $\rho((x_1,x_2))=a$ and $\rho((x_2,x_3))=b$ for some $x_1<x_2<x_3$ and $a\ne b$ can replace the Step activation function in our construction, regardless of its value outside of $(x_1,x_3)$. In other words, such $\rho$ can be optimization-friendly (i.e., having non-zero gradient) except for $(x_1,x_3)$ whose measure can be arbitrarily small.
>
> **Clarification on 'networks of width $m$.'** We thank the reviewer for pointing this out. We use the term '$\rho$ networks of width $m$' as the collection of all $\rho$ networks of width $m$ having a finite number of layers. We clarified this in the notation section (Section 2) of the revised draft.
>
> **Minimum number of layers for achieving minimum width.** Our upper bound proofs are based on concrete constructions of approximators, so we can compute the number of layers/parameters in our constructions. Our ReLU+Step network construction achieving the tight minimum width for approximating $f\in C([0,1]^{d_x},\mathbb R^{d_y})$ in $\varepsilon$ error requires $O\big((\omega_{f}^{-1}(\Omega(\varepsilon)))^{-d_x}+1/\varepsilon\big)$ parameters, where $\omega_{f}$ denotes the modulus of continuity of $f$. Likewise, Our ReLU network construction achieving the tight minimum width for approximating $f\in L^p([0,1]^{d_x},\mathbb R^{d_y})$ in $\varepsilon$ error requires $O\big((\omega_{g}^{-1}(\Omega(\varepsilon)))^{-d_x}+1/\varepsilon\big)$ parameters where $g$ is some continuous function satisfying $\|f-g\|_p\le\varepsilon/2$. More details can be found in the newly added Section 4.5 in the revised draft.
>
> **Implication for ReLU networks of fixed number of layers.**
> As we focused on identifying the minimum width of networks having an arbitrary number of layers, providing implications for two or three layer networks using our results is hard.
> However, if the number of layers is fixed (i.e., constant) but larger than two or three layers, our analysis of the required number of layers (our previous answer) can bound the error for approximating functions using networks of minimum width and fixed depth.
>
> We also note that networks of depth two or three have been actively studied in the past decades. For example, the classical universal approximation theorem states that two-layer networks can approximate any continuous/$L^p$ functions in arbitrary error by increasing width (e.g., Pinkus (1999)).
>
> **Comment on Table 1.** In Table 1, the row for Ours (Theorem 2) is stating that the minimum width $w_{\min}$ is 3 and this is strictly greater than $\max\{d_x+1, d_y\}$ because $d_x = 1$ and $d_y = 2$ in this case.

---

### Author Response · Authors · 2020-11-22
**Summary of revision**

Dear reviewers,

We express our deepest gratitude for your constructive feedback and valuable comments.

In response to the questions and concerns you raised, we have carefully revised and enhanced the draft with additional discussion on the required number of layers for our constructions (Section 4.5). The revisions made are marked with blue in the revised draft.

Thanks,

Authors

---

### Decision · Program_Chairs · 2021-01-07
**Final Decision**

**Decision:**

Accept (Spotlight)

**Comment:**

Two knowledgeable reviewers and one fairly confident reviewer were positive (7) about this submission. The authors' response clarified a few questions and comments from the initial reviews. The paper provides exact bounds that close the gap between lower and upper bounds, and that helps us understand these networks better. With the unanimously positive feedback, I am recommending the paper to be accepted.